# Composing graphical models with neural networks for structured representations and fast inference

**Matthew James Johnson**
Harvard University
mattjj@seas.harvard.edu

**David Duvenaud**
Harvard University
dduvenaud@seas.harvard.edu

**Alexander B. Wiltschko**
Harvard University, Twitter
awiltsch@fas.harvard.edu

**Sandeep R. Datta**
Harvard Medical School
srdatta@hms.harvard.edu

**Ryan P. Adams**
Harvard University, Twitter
rpa@seas.harvard.edu

## Abstract

We propose a general modeling and inference framework that combines the complementary strengths of probabilistic graphical models and deep learning methods. Our model family composes latent graphical models with neural network observation likelihoods. For inference, we use recognition networks to produce local evidence potentials, then combine them with the model distribution using efficient message-passing algorithms. All components are trained simultaneously with a single stochastic variational inference objective. We illustrate this framework by automatically segmenting and categorizing mouse behavior from raw depth video, and demonstrate several other example models.

## 1 Introduction

Modeling often has two goals: first, to learn a flexible representation of complex high-dimensional data, such as images or speech recordings, and second, to find structure that is interpretable and generalizes to new tasks. Probabilistic graphical models [1, 2] provide many tools to build structured representations, but often make rigid assumptions and may require significant feature engineering. Alternatively, deep learning methods allow flexible data representations to be learned automatically, but may not directly encode interpretable or tractable probabilistic structure. Here we develop a general modeling and inference framework that combines these complementary strengths.

Consider learning a generative model for video of a mouse. Learning interpretable representations for such data, and comparing them as the animal's genes are edited or its brain chemistry altered, gives useful behavioral phenotyping tools for neuroscience and for high-throughput drug discovery [3]. Even though each image is encoded by hundreds of pixels, the data lie near a low-dimensional nonlinear manifold. A useful generative model must not only learn this manifold but also provide an interpretable representation of the mouse's behavioral dynamics. A natural representation from ethology [3] is that the mouse's behavior is divided into brief, reused actions, such as darts, rears, and grooming bouts. Therefore an appropriate model might switch between discrete states, with each state representing the dynamics of a particular action. These two learning tasks — identifying an image manifold and a structured dynamics model — are complementary: we want to learn the image manifold in terms of coordinates in which the structured dynamics fit well. A similar challenge arises in speech [4], where high-dimensional spectrographic data lie near a low-dimensional manifold because they are generated by a physical system with relatively few degrees of freedom [5] but also include the discrete latent dynamical structure of phonemes, words, and grammar [6].

To address these challenges, we propose a new framework to design and learn models that couple nonlinear likelihoods with structured latent variable representations. Our approach uses graphical models for representing structured probability distributions while enabling fast exact inference subroutines, and uses ideas from variational autoencoders [7, 8] for learning not only the nonlinear

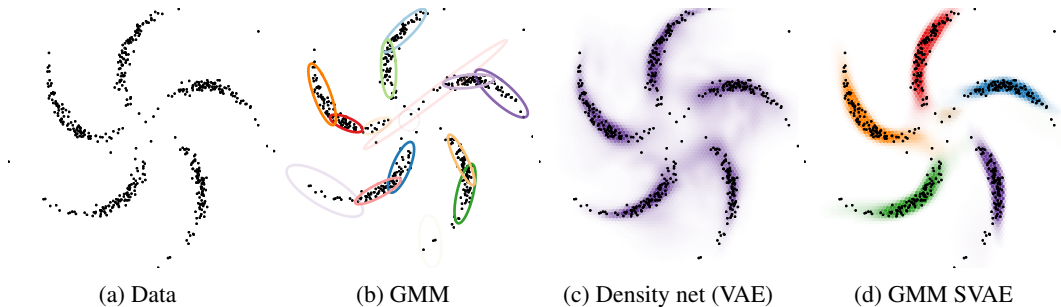

| (a) Data | (b) GMM | (c) Density net (VAE) | (d) GMM SVAE |

Figure 1: Comparison of generative models fit to spiral cluster data. See Section 2.1.

feature manifold but also bottom-up recognition networks to improve inference. Thus our method enables the combination of flexible deep learning feature models with structured Bayesian (and even nonparametric [9]) priors. Our approach yields a single variational inference objective in which all components of the model are learned simultaneously. Furthermore, we develop a scalable fitting algorithm that combines several advances in efficient inference, including stochastic variational inference [10], graphical model message passing [1], and backpropagation with the reparameterization trick [7]. Thus our algorithm can leverage conjugate exponential family structure where it exists to efficiently compute natural gradients with respect to some variational parameters, enabling effective second-order optimization [11], while using backpropagation to compute gradients with respect to all other parameters. We refer to our general approach as the structured variational autoencoder (SVAE).

## 2 Latent graphical models with neural net observations

In this paper we propose a broad family of models. Here we develop three specific examples.

### 2.1 Warped mixtures for arbitrary cluster shapes

One particularly natural structure used frequently in graphical models is the discrete mixture model. By fitting a discrete mixture model to data, we can discover natural clusters or units. These discrete structures are difficult to represent directly in neural network models.

Consider the problem of modeling the data $y = \{y_n\}_{n=1}^N$ shown in Fig. 1a. A standard approach to finding the clusters in data is to fit a Gaussian mixture model (GMM) with a conjugate prior:

$$\pi \sim \text{Dir}(\alpha), \quad (\mu_k, \Sigma_k) \overset{\text{iid}}{\sim} \text{NIW}(\lambda), \quad z_n \,|\, \pi \overset{\text{iid}}{\sim} \pi \quad y_n \,|\, z_n, \{(\mu_k, \Sigma_k)\}_{k=1}^K \overset{\text{iid}}{\sim} \mathcal{N}(\mu_{z_n}, \Sigma_{z_n}).$$

However, the fit GMM does not represent the natural clustering of the data (Fig. 1b). Its inflexible Gaussian observation model limits its ability to parsimoniously fit the data and their natural semantics.

Instead of using a GMM, a more flexible alternative would be a neural network density model:

$$\gamma \sim p(\gamma) \qquad x_n \overset{\text{iid}}{\sim} \mathcal{N}(0, I), \qquad y_n \,|\, x_n, \gamma \overset{\text{iid}}{\sim} \mathcal{N}(\mu(x_n; \gamma), \Sigma(x_n; \gamma)), \tag{1}$$

where $\mu(x_n; \gamma)$ and $\Sigma(x_n; \gamma)$ depend on $x_n$ through some smooth parametric function, such as multilayer perceptron (MLP), and where $p(\gamma)$ is a Gaussian prior [12]. This model fits the data density well (Fig. 1c) but does not explicitly represent discrete mixture components, which might provide insights into the data or natural units for generalization. See Fig. 2a for a graphical model.

By composing a latent GMM with nonlinear observations, we can combine the modeling strengths of both [13], learning both discrete clusters along with non-Gaussian cluster shapes:

$$\pi \sim \text{Dir}(\alpha), \quad (\mu_k, \Sigma_k) \overset{\text{iid}}{\sim} \text{NIW}(\lambda), \quad \gamma \sim p(\gamma)$$

$$z_n \,|\, \pi \overset{\text{iid}}{\sim} \pi \quad x_n \overset{\text{iid}}{\sim} \mathcal{N}(\mu^{(z_n)}, \Sigma^{(z_n)}), \quad y_n \,|\, x_n, \gamma \overset{\text{iid}}{\sim} \mathcal{N}(\mu(x_n; \gamma), \Sigma(x_n; \gamma)).$$

This combination of flexibility and structure is shown in Fig. 1d. See Fig. 2b for a graphical model.

### 2.2 Latent linear dynamical systems for modeling video

Now we consider a harder problem: generatively modeling video. Since a video is a sequence of image frames, a natural place to start is with a model for images. Kingma et al. [7] shows that the

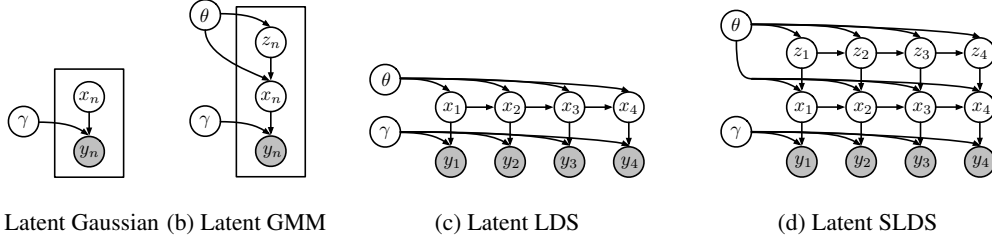

| (a) Latent Gaussian | (b) Latent GMM | (c) Latent LDS | (d) Latent SLDS |

Figure 2: Generative graphical models discussed in Section 2.

density network of Eq. (1) can accurately represent a dataset of high-dimensional images $\{y_n\}_{n=1}^N$ in terms of the low-dimensional latent variables $\{x_n\}_{n=1}^N$, each with independent Gaussian distributions.

To extend this image model into a model for videos, we can introduce dependence through time between the latent Gaussian samples $\{x_n\}_{n=1}^N$. For instance, we can make each latent variable $x_{n+1}$ depend on the previous latent variable $x_n$ through a Gaussian linear dynamical system, writing

$$x_{n+1} = Ax_n + Bu_n, \qquad u_n \overset{\text{iid}}{\sim} \mathcal{N}(0, I), \qquad A, B \in \mathbb{R}^{m \times m},$$

where the matrices $A$ and $B$ have a conjugate prior. This model has low-dimensional latent states and dynamics as well as a rich nonlinear generative model of images. In addition, the timescales of the dynamics are represented directly in the eigenvalue spectrum of $A$, providing both interpretability and a natural way to encode prior information. See Fig. 2c for a graphical model.

## 2.3   Latent switching linear dynamical systems for parsing behavior from video

As a final example that combines both time series structure and discrete latent units, consider again the behavioral phenotyping problem described in Section 1. Drawing on graphical modeling tools, we can construct a latent switching linear dynamical system (SLDS) [14] to represent the data in terms of continuous latent states that evolve according to a discrete library of linear dynamics, and drawing on deep learning methods we can generate video frames with a neural network image model.

At each time $n \in \{1, 2, \ldots, N\}$ there is a discrete-valued latent state $z_n \in \{1, 2, \ldots, K\}$ that evolves according to Markovian dynamics. The discrete state indexes a set of linear dynamical parameters, and the continuous-valued latent state $x_n \in \mathbb{R}^m$ evolves according to the corresponding dynamics,

$$z_{n+1} \,|\, z_n, \pi \sim \pi_{z_n}, \qquad x_{n+1} = A_{z_n} x_n + B_{z_n} u_n, \qquad u_n \overset{\text{iid}}{\sim} \mathcal{N}(0, I),$$

where $\pi = \{\pi_k\}_{k=1}^K$ denotes the Markov transition matrix and $\pi_k \in \mathbb{R}_+^K$ is its $k$th row. We use the same neural net observation model as in Section 2.2. This SLDS model combines both continuous and discrete latent variables with rich nonlinear observations. See Fig. 2d for a graphical model.

## 3   Structured mean field inference and recognition networks

Why aren't such rich hybrid models used more frequently? The main difficulty with combining rich latent variable structure and flexible likelihoods is inference. The most efficient inference algorithms used in graphical models, like structured mean field and message passing, depend on conjugate exponential family likelihoods to preserve tractable structure. When the observations are more general, like neural network models, inference must either fall back to general algorithms that do not exploit the model structure or else rely on bespoke algorithms developed for one model at a time.

In this section, we review inference ideas from conjugate exponential family probabilistic graphical models and variational autoencoders, which we combine and generalize in the next section.

### 3.1   Inference in graphical models with conjugacy structure

Graphical models and exponential families provide many algorithmic tools for efficient inference [15]. Given an exponential family latent variable model, when the observation model is a conjugate exponential family, the conditional distributions stay in the same exponential families as in the prior and hence allow for the same efficient inference algorithms.

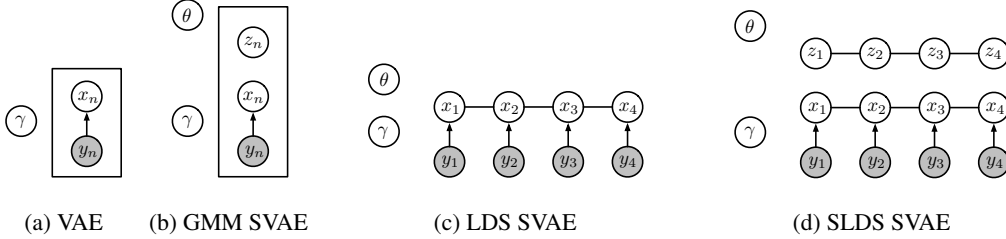

|     (a) VAE     |   (b) GMM SVAE   |   (c) LDS SVAE   |   (d) SLDS SVAE   |

Figure 3: Variational families and recognition networks for the VAE [7] and three SVAE examples.

For example, consider learning a Gaussian linear dynamical system model with linear Gaussian observations. The generative model for latent states $x = \{x_n\}_{n=1}^N$ and observations $y = \{y_n\}_{n=1}^N$ is

$$x_n = Ax_{n-1} + Bu_{n-1}, \qquad u_n \overset{\text{iid}}{\sim} \mathcal{N}(0, I), \qquad y_n = Cx_n + Dv_n, \qquad v_n \overset{\text{iid}}{\sim} \mathcal{N}(0, I),$$

given parameters $\theta = (A, B, C, D)$ with a conjugate prior $p(\theta)$. To approximate the posterior $p(\theta, x \,|\, y)$, consider the mean field family $q(\theta)q(x)$ and the variational inference objective

$$\mathcal{L}[\, q(\theta)q(x) \,] = \mathbb{E}_{q(\theta)q(x)} \left[ \log \frac{p(\theta)p(x \,|\, \theta)p(y \,|\, x, \theta)}{q(\theta)q(x)} \right], \tag{2}$$

where we can optimize the variational family $q(\theta)q(x)$ to approximate the posterior $p(\theta, x \,|\, y)$ by maximizing Eq. (2). Because the observation model $p(y \,|\, x, \theta)$ is conjugate to the latent variable model $p(x \,|\, \theta)$, for any fixed $q(\theta)$ the optimal factor $q^*(x) \triangleq \arg\max_{q(x)} \mathcal{L}[\, q(\theta)q(x) \,]$ is itself a Gaussian linear dynamical system with parameters that are simple functions of the expected statistics of $q(\theta)$ and the data $y$. As a result, for fixed $q(\theta)$ we can easily compute $q^*(x)$ and use message passing algorithms to perform exact inference in it. However, when the observation model is not conjugate to the latent variable model, these algorithmically exploitable structures break down.

### 3.2 Recognition networks in variational autoencoders

The variational autoencoder (VAE) [7] handles general non-conjugate observation models by introducing recognition networks. For example, when a Gaussian latent variable model $p(x)$ is paired with a general nonlinear observation model $p(y \,|\, x, \gamma)$, the posterior $p(x \,|\, y, \gamma)$ is non-Gaussian, and it is difficult to compute an optimal Gaussian approximation. The VAE instead learns to directly output a suboptimal Gaussian factor $q(x \,|\, y)$ by fitting a parametric map from data $y$ to a mean and covariance, $\mu(y; \phi)$ and $\Sigma(y; \phi)$, such as an MLP with parameters $\phi$. By optimizing over $\phi$, the VAE effectively learns how to condition on non-conjugate observations $y$ and produce a good approximating factor.

## 4 Structured variational autoencoders

We can combine the tractability of conjugate graphical model inference with the flexibility of variational autoencoders. The main idea is to use a conditional random field (CRF) variational family. We learn recognition networks that output conjugate graphical model potentials instead of outputting the complete variational distribution's parameters directly. These potentials are then used in graphical model inference algorithms in place of the non-conjugate observation likelihoods.

The SVAE algorithm computes stochastic gradients of a mean field variational inference objective. It can be viewed as a generalization both of the natural gradient SVI algorithm for conditionally conjugate models [10] and of the AEVB algorithm for variational autoencoders [7]. Intuitively, it proceeds by sampling a data minibatch, applying the recognition model to compute graphical model potentials, and using graphical model inference algorithms to compute the variational factor, combining the evidence from the potentials with the prior structure in the model. This variational factor is then used to compute gradients of the mean field objective. See Fig. 3 for graphical models of the variational families with recognition networks for the models developed in Section 2.

In this section, we outline the SVAE model class more formally, write the mean field variational inference objective, and show how to efficiently compute unbiased stochastic estimates of its gradients. The resulting algorithm for computing gradients of the mean field objective, shown in Algorithm 1, is

---

**Algorithm 1** Estimate SVAE lower bound and its gradients

---

**Input:** Variational parameters $(\eta_\theta, \eta_\gamma, \phi)$, data sample $y$
  **function** SVAEGRADIENTS$(\eta_\theta, \eta_\gamma, \phi, y)$
    $\psi \leftarrow r(y_n; \phi)$                                                  $\triangleright$ Get evidence potentials
    $(\hat{x}, \bar{t}_x, \text{KL}^{\text{local}}) \leftarrow \text{PGMINFERENCE}(\eta_\theta, \psi)$          $\triangleright$ Combine evidence with prior
    $\hat{\gamma} \sim q(\gamma)$                                                     $\triangleright$ Sample observation parameters
    $\mathcal{L} \leftarrow N \log p(y \,|\, \hat{x}, \hat{\gamma}) - N \, \text{KL}^{\text{local}} - \text{KL}(q(\theta)q(\gamma)\|p(\theta)p(\gamma))$    $\triangleright$ Estimate variational bound
    $\widetilde{\nabla}_{\eta_\theta} \mathcal{L} \leftarrow \eta_\theta^0 - \eta_\theta + N(\bar{t}_x, 1) + N(\nabla_{\eta_x} \log p(y \,|\, \hat{x}, \hat{\gamma}), 0)$    $\triangleright$ Compute natural gradient
    **return** lower bound $\mathcal{L}$, natural gradient $\widetilde{\nabla}_{\eta_\theta} \mathcal{L}$, gradients $\nabla_{\eta_\gamma, \phi} \mathcal{L}$
  **function** PGMINFERENCE$(\eta_\theta, \psi)$
    $q^*(x) \leftarrow \text{OPTIMIZELOCALFACTORS}(\eta_\theta, \psi)$              $\triangleright$ Fast message-passing inference
    **return** sample $\hat{x} \sim q^*(x)$, statistics $\mathbb{E}_{q^*(x)} t_x(x)$, divergence $\mathbb{E}_{q(\theta)} \text{KL}(q^*(x)\|p(x \,|\, \theta))$

---

simple and efficient and can be readily applied to a variety of learning problems and graphical model structures. See the supplementals for details and proofs.

## 4.1 SVAE model class

To set up notation for a general SVAE, we first define a conjugate pair of exponential family densities on global latent variables $\theta$ and local latent variables $x = \{x_n\}_{n=1}^N$. Let $p(x \,|\, \theta)$ be an exponential family and let $p(\theta)$ be its corresponding natural exponential family conjugate prior, writing

$$p(\theta) = \exp\left\{\langle \eta_\theta^0, t_\theta(\theta) \rangle - \log Z_\theta(\eta_\theta^0)\right\},$$

$$p(x \,|\, \theta) = \exp\left\{\langle \eta_x^0(\theta), t_x(x) \rangle - \log Z_x(\eta_x^0(\theta))\right\} = \exp\left\{\langle t_\theta(\theta), (t_x(x), 1) \rangle\right\},$$

where we used exponential family conjugacy to write $t_\theta(\theta) = \left(\eta_x^0(\theta), -\log Z_x(\eta_x^0(\theta))\right)$. The local latent variables $x$ could have additional structure, like including both discrete and continuous latent variables or tractable graph structure, but here we keep the notation simple.

Next, we define a general likelihood function. Let $p(y \,|\, x, \gamma)$ be a general family of densities and let $p(\gamma)$ be an exponential family prior on its parameters. For example, each observation $y_n$ may depend on the latent value $x_n$ through an MLP, as in the density network model of Section 2. This generic non-conjugate observation model provides modeling flexibility, yet the SVAE can still leverage conjugate exponential family structure in inference, as we show next.

## 4.2 Stochastic variational inference algorithm

Though the general observation model $p(y \,|\, x, \gamma)$ means that conjugate updates and natural gradient SVI [10] cannot be directly applied, we show that by generalizing the recognition network idea we can still approximately optimize out the local variational factors leveraging conjugacy structure.

For fixed $y$, consider the mean field family $q(\theta)q(\gamma)q(x)$ and the variational inference objective

$$\mathcal{L}[\, q(\theta)q(\gamma)q(x) \,] \triangleq \mathbb{E}_{q(\theta)q(\gamma)q(x)}\left[\log \frac{p(\theta)p(\gamma)p(x \,|\, \theta)p(y \,|\, x, \gamma)}{q(\theta)q(\gamma)q(x)}\right]. \tag{3}$$

Without loss of generality we can take the global factor $q(\theta)$ to be in the same exponential family as the prior $p(\theta)$, and we denote its natural parameters by $\eta_\theta$. We restrict $q(\gamma)$ to be in the same exponential family as $p(\gamma)$ with natural parameters $\eta_\gamma$. Finally, we restrict $q(x)$ to be in the same exponential family as $p(x \,|\, \theta)$, writing its natural parameter as $\eta_x$. Using these explicit variational parameters, we write the mean field variational inference objective in Eq. (3) as $\mathcal{L}(\eta_\theta, \eta_\gamma, \eta_x)$.

To perform efficient optimization of the objective $\mathcal{L}(\eta_\theta, \eta_\gamma, \eta_x)$, we consider choosing the variational parameter $\eta_x$ as a function of the other parameters $\eta_\theta$ and $\eta_\gamma$. One natural choice is to set $\eta_x$ to be a local partial optimizer of $\mathcal{L}$. However, without conjugacy structure finding a local partial optimizer may be computationally expensive for general densities $p(y \,|\, x, \gamma)$, and in the large data setting this expensive optimization would have to be performed for each stochastic gradient update. Instead, we choose $\eta_x$ by optimizing over a surrogate objective $\widehat{\mathcal{L}}$ with conjugacy structure, given by

$$\widehat{\mathcal{L}}(\eta_\theta, \eta_x, \phi) \triangleq \mathbb{E}_{q(\theta)q(x)}\left[\log \frac{p(\theta)p(x \,|\, \theta)\exp\{\psi(x; y, \phi)\}}{q(\theta)q(x)}\right], \quad \psi(x; y, \phi) \triangleq \langle r(y; \phi), \, t_x(x) \rangle,$$

where $\{r(y;\phi)\}_{\phi \in \mathbb{R}^m}$ is some parameterized class of functions that serves as the recognition model. Note that the potentials $\psi(x; y, \phi)$ have a form conjugate to the exponential family $p(x \mid \theta)$. We define $\eta_x^*(\eta_\theta, \phi)$ to be a local partial optimizer of $\widehat{\mathcal{L}}$ along with the corresponding factor $q^*(x)$,

$$\eta_x^*(\eta_\theta, \phi) \triangleq \arg\min_{\eta_x} \widehat{\mathcal{L}}(\eta_\theta, \eta_x, \phi), \qquad q^*(x) = \exp\left\{ \langle \eta_x^*(\eta_\theta, \phi), \, t_x(x) \rangle - \log Z_x(\eta_x^*(\eta_\theta, \phi)) \right\}.$$

As with the variational autoencoder of Section 3.2, the resulting variational factor $q^*(x)$ is suboptimal for the variational objective $\mathcal{L}$. However, because the surrogate objective has the same form as a variational inference objective for a conjugate observation model, the factor $q^*(x)$ not only is easy to compute but also inherits exponential family and graphical model structure for tractable inference.

Given this choice of $\eta_x^*(\eta_\theta, \phi)$, the SVAE objective is $\mathcal{L}_{\mathrm{SVAE}}(\eta_\theta, \eta_\gamma, \phi) \triangleq \mathcal{L}(\eta_\theta, \eta_\gamma, \eta_x^*(\eta_\theta, \phi))$. This objective is a lower bound for the variational inference objective Eq. (3) in the following sense.

**Proposition 4.1** (The SVAE objective lower-bounds the mean field objective)
*The SVAE objective function $\mathcal{L}_{\mathrm{SVAE}}$ lower-bounds the mean field objective $\mathcal{L}$ in the sense that*

$$\max_{q(x)} \mathcal{L}[\, q(\theta)q(\gamma)q(x) \,] \geq \max_{\eta_x} \mathcal{L}(\eta_\theta, \eta_\gamma, \eta_x) \geq \mathcal{L}_{\mathrm{SVAE}}(\eta_\theta, \eta_\gamma, \phi) \quad \forall \phi \in \mathbb{R}^m,$$

*for any parameterized function class $\{r(y;\phi)\}_{\phi \in \mathbb{R}^m}$. Furthermore, if there is some $\phi^* \in \mathbb{R}^m$ such that $\psi(x; y, \phi^*) = \mathbb{E}_{q(\gamma)} \log p(y \mid x, \gamma)$, then the bound can be made tight in the sense that*

$$\max_{q(x)} \mathcal{L}[\, q(\theta)q(\gamma)q(x) \,] = \max_{\eta_x} \mathcal{L}(\eta_\theta, \eta_\gamma, \eta_x) = \max_{\phi} \mathcal{L}_{\mathrm{SVAE}}(\eta_\theta, \eta_\gamma, \phi).$$

Thus by using gradient-based optimization to maximize $\mathcal{L}_{\mathrm{SVAE}}(\eta_\theta, \eta_\gamma, \phi)$ we are maximizing a lower bound on the model log evidence $\log p(y)$. In particular, by optimizing over $\phi$ we are effectively learning how to condition on observations so as to best approximate the posterior while maintaining conjugacy structure. Furthermore, to provide the best lower bound we may choose the recognition model function class $\{r(y;\phi)\}_{\phi \in \mathbb{R}^m}$ to be as rich as possible.

Choosing $\eta_x^*(\eta_\theta, \phi)$ to be a local partial optimizer of $\widehat{\mathcal{L}}$ provides two computational advantages. First, it allows $\eta_x^*(\eta_\theta, \phi)$ and expectations with respect to $q^*(x)$ to be computed efficiently by exploiting exponential family graphical model structure. Second, it provides a simple expression for an unbiased estimate of the natural gradient with respect to the latent model parameters, as we summarize next.

**Proposition 4.2** (Natural gradient of the SVAE objective)
*The natural gradient of the SVAE objective $\mathcal{L}_{\mathrm{SVAE}}$ with respect to $\eta_\theta$ is*

$$\widetilde{\nabla}_{\eta_\theta} \mathcal{L}_{\mathrm{SVAE}}(\eta_\theta, \eta_\gamma, \phi) = \left( \eta_\theta^0 + \mathbb{E}_{q^*(x)} \left[ (t_x(x), 1) \right] - \eta_\theta \right) + (\nabla_{\eta_x} \mathcal{L}(\eta_\theta, \eta_\gamma, \eta_x^*(\eta_\theta, \phi)), 0). \tag{4}$$

Note that the first term in Eq. (4) is the same as the expression for the natural gradient in SVI for conjugate models [10], while a stochastic estimate of the second term is computed automatically as part of the backward pass for computing the gradients with respect to the other parameters, as described next. Thus we have an expression for the natural gradient with respect to the latent model's parameters that is almost as simple as the one for conjugate models and just as easy to compute. Natural gradients are invariant to smooth invertible reparameterizations of the variational family [16, 17] and provide effective second-order optimization updates [18, 11].

The gradients of the objective with respect to the other variational parameters, namely $\nabla_{\eta_\gamma} \mathcal{L}_{\mathrm{SVAE}}(\eta_\theta, \eta_\gamma, \phi)$ and $\nabla_\phi \mathcal{L}_{\mathrm{SVAE}}(\eta_\theta, \eta_\gamma, \phi)$, can be computed using the reparameterization trick. To isolate the terms that require the reparameterization trick, we rearrange the objective as

$$\mathcal{L}_{\mathrm{SVAE}}(\eta_\theta, \eta_\gamma, \phi) = \mathbb{E}_{q(\gamma)q^*(x)} \log p(y \mid x, \gamma) - \mathrm{KL}(q(\theta)q^*(x) \,\|\, p(\theta, x)) - \mathrm{KL}(q(\gamma) \,\|\, p(\gamma)).$$

The KL divergence terms are between members of the same tractable exponential families. An unbiased estimate of the first term can be computed by sampling $\hat{x} \sim q^*(x)$ and $\hat{\gamma} \sim q(\gamma)$ and computing $\nabla_{\eta_\gamma, \phi} \log p(y \mid \hat{x}, \hat{\gamma})$ with automatic differentiation. Note that the second term in Eq. (4) is automatically computed as part of the chain rule in computing $\nabla_\phi \log p(y \mid \hat{x}, \hat{\gamma})$.

## 5   Related work

In addition to the papers already referenced, there are several recent papers to which this work is related. The two papers closest to this work are Krishnan et al. [19] and Archer et al. [20].

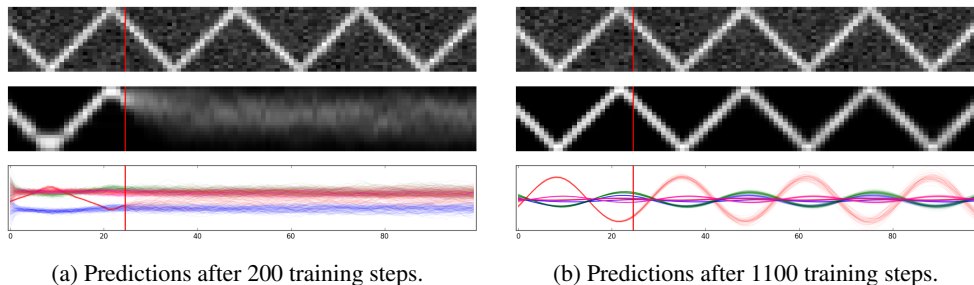

(a) Predictions after 200 training steps.　　　(b) Predictions after 1100 training steps.

Figure 4: Predictions from an LDS SVAE fit to 1D dot image data at two stages of training. The top panel shows an example sequence with time on the horizontal axis. The middle panel shows the noiseless predictions given data up to the vertical line, while the bottom panel shows the latent states.

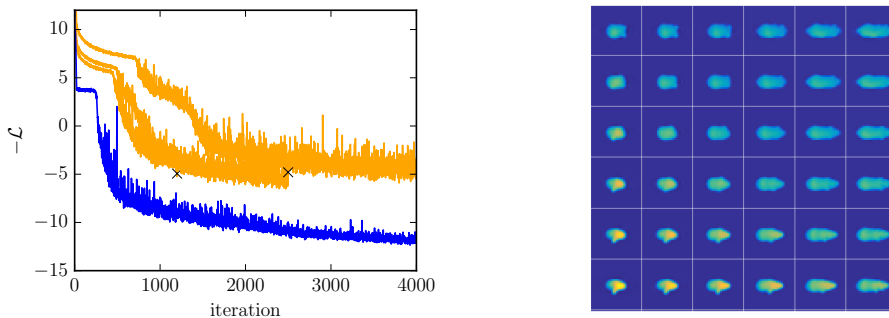

(a) Natural (blue) and standard (orange) gradient updates.　　　(b) Subspace of learned observation model.

Figure 5: Experimental results from LDS SVAE models on synthetic data and real mouse data.

In Krishnan et al. [19] the authors consider combining variational autoencoders with continuous state-space models, emphasizing the relationship to linear dynamical systems (also called Kalman filter models). They primarily focus on nonlinear dynamics and an RNN-based variational family, as well as allowing control inputs. However, the approach does not extend to general graphical models or discrete latent variables. It also does not leverage natural gradients or exact inference subroutines.

In Archer et al. [20] the authors also consider the problem of variational inference in general continuous state space models but focus on using a structured Gaussian variational family without considering parameter learning. As with Krishnan et al. [19], this approach does not include discrete latent variables (or any latent variables other than the continuous states). However, the method they develop could be used with an SVAE to handle inference with nonlinear dynamics.

In addition, both Gregor et al. [21] and Chung et al. [22] extend the variational autoencoder framework to sequential models, though they focus on RNNs rather than probabilistic graphical models.

## 6 Experiments

We apply the SVAE to both synthetic and real data and demonstrate its ability to learn feature representations and latent structure. Code is available at github.com/mattjj/svae.

### 6.1 LDS SVAE for modeling synthetic data

Consider a sequence of 1D images representing a dot bouncing from one side of the image to the other, as shown at the top of Fig. 4. We use an LDS SVAE to find a low-dimensional latent state space representation along with a nonlinear image model. The model is able to represent the image accurately and to make long-term predictions with uncertainty. See supplementals for details.

This experiment also demonstrates the optimization advantages that can be provided by the natural gradient updates. In Fig. 5a we compare natural gradient updates with standard gradient updates at three different learning rates. The natural gradient algorithm not only learns much faster but also is less dependent on parameterization details: while the natural gradient update used an untuned

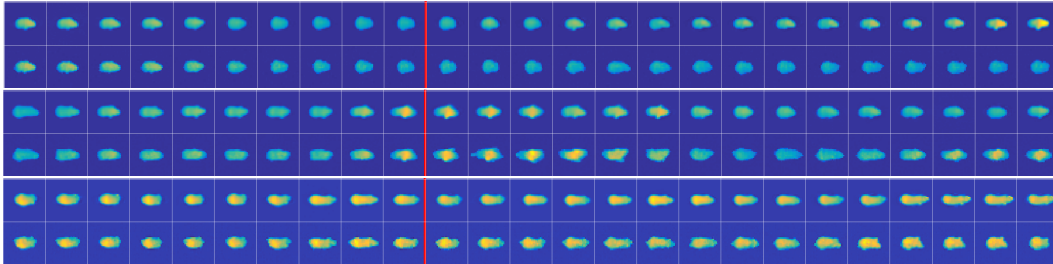

Figure 6: Predictions from an LDS SVAE fit to depth video. In each panel, the top is a sampled prediction and the bottom is real data. The model is conditioned on observations to the left of the line.

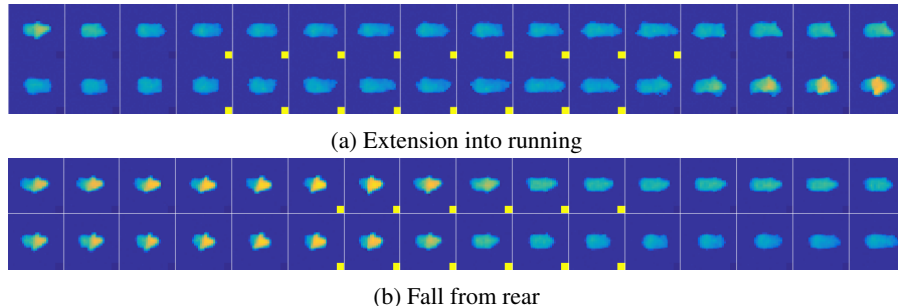

(a) Extension into running

(b) Fall from rear

Figure 7: Examples of behavior states inferred from depth video. Each frame sequence is padded on both sides, with a square in the lower-right of a frame depicting when the state is the most probable.

stepsize of 0.1, the standard gradient dynamics at step sizes of both 0.1 and 0.05 resulted in some matrix parameters to be updated to indefinite values.

## 6.2 LDS SVAE for modeling video

We also apply an LDS SVAE to model depth video recordings of mouse behavior. We use the dataset from Wiltschko et al. [3] in which a mouse is recorded from above using a Microsoft Kinect. We used a subset consisting of 8 recordings, each of a distinct mouse, 20 minutes long at 30 frames per second, for a total of 288000 video fames downsampled to $30 \times 30$ pixels.

We use MLP observation and recognition models with two hidden layers of 200 units each and a 10D latent space. Fig. 5b shows images corresponding to a regular grid on a random 2D subspace of the latent space, illustrating that the learned image manifold accurately captures smooth variation in the mouse's body pose. Fig. 6 shows predictions from the model paired with real data.

## 6.3 SLDS SVAE for parsing behavior

Finally, because the LDS SVAE can accurately represent the depth video over short timescales, we apply the latent switching linear dynamical system (SLDS) model to discover the natural units of behavior. Fig. 7 shows some of the discrete states that arise from fitting an SLDS SVAE with 30 discrete states to the depth video data. The discrete states that emerge show a natural clustering of short-timescale patterns into behavioral units. See the supplementals for more.

## 7 Conclusion

Structured variational autoencoders provide a general framework that combines some of the strengths of probabilistic graphical models and deep learning methods. In particular, they use graphical models both to give models rich latent representations and to enable fast variational inference with CRF structured approximating distributions. To complement these structured representations, SVAEs use neural networks to produce not only flexible nonlinear observation models but also fast recognition networks that map observations to conjugate graphical model potentials.

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
