[Supplementary Material]

# A Optimization

In this section we fix our notation for gradients and establish some basic definitions and results that we use in the sequel.

## A.1 Gradient notation

We follow the notation in Bertsekas [1, A.5]. In particular, if $f : \mathbb{R}^n \to \mathbb{R}^m$ is a continuously differentiable function, we define the gradient matrix of $f$, denoted $\nabla f(x)$, to be the $n \times m$ matrix in which the $i$th column is the gradient $\nabla f_i(x)$ of $f_i$, the $i$th coordinate function of $f$, for $i = 1, 2, \ldots, m$. That is,

$$\nabla f(x) = [\nabla f_1(x) \quad \cdots \quad \nabla f_m(x)].$$

The transpose of $\nabla f$ is the Jacobian matrix of $f$, in which the $ij$th entry is the function $\partial f_i / \partial x_j$.

If $f : \mathbb{R}^n \to \mathbb{R}$ is continuously differentiable with continuously differentiable partial derivatives, then we define the Hessian matrix of $f$, denoted $\nabla^2 f$, to be the matrix in which the $ij$th entry is the function $\partial^2 f / \partial x_i \partial x_j$.

Finally, if $f : \mathbb{R}^n \times \mathbb{R}^m \to \mathbb{R}$ is a function of $(x, y)$ with $x \in R^n$ and $y \in \mathbb{R}^m$, we write

$$\nabla_x f(x, y) = \begin{pmatrix} \frac{\partial f(x,y)}{\partial x_1} \\ \vdots \\ \frac{\partial f(x,y)}{\partial x_m} \end{pmatrix}, \qquad \nabla_y f(x, y) = \begin{pmatrix} \frac{\partial f(x,y)}{\partial y_1} \\ \vdots \\ \frac{\partial f(x,y)}{\partial y_n} \end{pmatrix}$$

$$\nabla^2_{xx} f(x, y) = \left( \frac{\partial^2 f(x, y)}{\partial x_i \partial x_j} \right), \qquad \nabla^2_{yy} f(x, y) = \left( \frac{\partial^2 f(x, y)}{\partial y_i \partial y_j} \right),$$

$$\nabla^2_{xy} f(x, y) = \left( \frac{\partial^2 f(x, y)}{\partial x_i \partial y_j} \right).$$

## A.2 Local and partial optimizers

In this section we state the definitions of local partial optimizer and necessary conditions for optimality that we use in the sequel.

**Definition A.1** (Partial optimizer, local partial optimizer)
*Let $f : \mathbb{R}^n \times \mathbb{R}^m \to \mathbb{R}$ be an objective function to be maximized. For a fixed $x \in \mathbb{R}^n$, we call a point $y^* \in \mathbb{R}^m$ an* unconstrained partial optimizer *of $f$ given $x$ if*

$$f(x, y) \leq f(x, y^*) \quad \forall y \in \mathbb{R}^m$$

*and we call $y^*$ an* unconstrained local partial optimizer *of $f$ given $x$ if there exists an $\epsilon > 0$ such that*

$$f(x, y) \leq f(x, y^*) \quad \forall y \text{ with } \|y - y^*\| < \epsilon,$$

*where $\| \cdot \|$ is any vector norm.*

**Proposition A.2** (Necessary conditions for optimality, Prop. 3.1.1 of Bertsekas [1])
*Let $f : \mathbb{R}^n \times \mathbb{R}^m \to \mathbb{R}$ be continuously differentiable. For fixed $x \in \mathbb{R}^n$ if $y^* \in \mathbb{R}^m$ is an unconstrained local partial optimizer for $f$ given $x$ then*

$$\nabla_y f(x, y^*) = 0.$$

*If instead $x$ and $y$ are subject to the constraints $h(x, y) = 0$ for some continuously differentiable $h : \mathbb{R}^n \times \mathbb{R}^m \to \mathbb{R}^m$ and $y^*$ is a constrained local partial optimizer for $f$ given $x$ with the regularity condition that $\nabla_y h(x, y^*)$ is full rank, then there exists a Lagrange multiplier $\lambda^* \in \mathbb{R}^m$ such that*

$$\nabla_y f(x, y^*) + \nabla_y h(x, y^*)\lambda^* = 0,$$

*and hence the cost gradient $\nabla_y f(x, y^*)$ is orthogonal to the first-order feasible variations in $y$ given by the null space of $\nabla_y h(x, y^*)^\mathsf{T}$.*

Note that the regularity condition on the constraints is not needed if the constraints are linear [1, Prop. 3.3.7].

For a continuously differentiable function $f : \mathbb{R}^n \to \mathbb{R}$, we say $x^*$ is a stationary point of $f$ if $\nabla f(x^*) = 0$. For general unconstrained smooth optimization, the limit points of gradient-based algorithms are guaranteed only to be stationary points of the objective, not necessarily local optima. Block coordinate ascent methods, when available, provide slightly stronger guarantees: not only is every limit point a stationary point of the objective, in addition each coordinate block is a partial optimizer of the objective. Note that the objective functions we consider maximizing in the following are bounded above.

### A.3  Partial optimization and the Implicit Function Theorem

Let $f : \mathbb{R}^n \times \mathbb{R}^m \to \mathbb{R}$ be a scalar-valued objective function of two unconstrained arguments $x \in \mathbb{R}^n$ and $y \in \mathbb{R}^m$, and let $y^* : \mathbb{R}^n \to \mathbb{R}^m$ be some function that assigns to each $x \in \mathbb{R}^n$ a value $y^*(x) \in \mathbb{R}^m$. Define the composite function $g : \mathbb{R}^n \to \mathbb{R}$ as

$$g(x) \triangleq f(x, y^*(x))$$

and using the chain rule write its gradient as

$$\nabla g(x) = \nabla_x f(x, y^*(x)) + \nabla y^*(x) \nabla_y f(x, y^*(x)). \tag{1}$$

One choice of the function $y^*(x)$ is to partially optimize $f$ for any fixed value of $x$. For example, assuming that $\arg\max_y f(x, y)$ is nonempty for every $x \in \mathbb{R}^n$, we could choose $y^*$ to satisfy $y^*(x) \in \arg\max_y f(x, y)$, so that $g(x) = \max_y f(x, y)$.[1] Similarly, if $y^*(x)$ is chosen so that $\nabla_y f(x, y^*(x)) = 0$, which is satisfied when $y^*(x)$ is an unconstrained local partial optimizer for $f$ given $x$, then the expression in Eq. (1) can be simplified as in the following proposition.

**Proposition A.3** (Gradients of locally partially optimized objectives)
*Let $f : \mathbb{R}^n \times \mathbb{R}^m \to \mathbb{R}$ be continuously differentiable, let $y^*$ be a local partial optimizer of $f$ given $x$ such that $y^*(x)$ is differentiable, and define $g(x) = f(x, y^*(x))$. Then*

$$\nabla g(x) = \nabla_x f(x, y^*(x)).$$

*Proof.* If $y^*$ is an unconstrained local partial optimizer of $f$ given $x$ then it satisfies $\nabla_y f(x, y^*) = 0$, and if $y^*$ is a regularly-constrained local partial optimizer then the feasible variation $\nabla y^*(x)$ is orthogonal to the cost gradient $\nabla_y f(x, y^*)$. In both cases the second term in the expression for $\nabla g(x)$ in Eq. (1) is zero. □

In general, when $y^*(x)$ is not a stationary point of $f(x, \cdot)$, to evaluate the gradient $\nabla g(x)$ we need to evaluate $\nabla y^*(x)$ in Eq. (1). However, this term may be difficult to compute directly. The function $y^*(x)$ may arise implicitly from some system of equations of the form $h(x, y) = 0$ for some continuously differentiable function $h : \mathbb{R}^n \times \mathbb{R}^m \to \mathbb{R}^m$. For example, the value of $y$ may be computed from $x$ and $h$ using a black-box iterative numerical algorithm. However, the Implicit Function Theorem provides another means to compute $\nabla y^*(x)$ using only the derivatives of $h$ and the value of $y^*(x)$.

**Proposition A.4** (Implicit Function Theorem, Prop. A.25 of Bertsekas [1])
*Let $h : \mathbb{R}^n \times \mathbb{R}^m \to \mathbb{R}^m$ be a function and $\bar{x} \in \mathbb{R}^n$ and $\bar{y} \in \mathbb{R}^m$ be points such that*

1. *$h(\bar{x}, \bar{y}) = 0$*

2. *$h$ is continuous and has a continuous nonsingular gradient matrix $\nabla_y h(x, y)$ in an open set containing $(\bar{x}, \bar{y})$.*

*Then there exist open sets $S_{\bar{x}} \subseteq \mathbb{R}^n$ and $S_{\bar{y}} \subseteq \mathbb{R}^m$ containing $\bar{x}$ and $\bar{y}$, respectively, and a continuous function $y^* : S_{\bar{x}} \to S_{\bar{y}}$ such that $\bar{y} = y^*(\bar{x})$ and $h(x, y^*(x)) = 0$ for all $x \in S_{\bar{x}}$. The function $y^*$ is*

*unique in the sense that if $x \in S_{\bar{x}}$, $y \in S_{\bar{y}}$, and $h(x, y) = 0$, then $y = y^*(x)$. Furthermore, if for some $p > 0$, $h$ is $p$ times continuously differentiable, the same is true for $y^*$, and we have*

$$\nabla y^*(x) = -\nabla_x h\left(x, y^*(x)\right) \left(\nabla_y h\left(x, y^*(x)\right)\right)^{-1}, \qquad \forall\, x \in S_{\bar{x}}.$$

As a special case, the equations $h(x, y) = 0$ may be the first-order stationary conditions of another unconstrained optimization problem. That is, the value of $y$ may be chosen by locally partially optimizing the value of $u(x, y)$ for a function $u : \mathbb{R}^n \times \mathbb{R}^m \to \mathbb{R}$ with no constraints on $y$, leading to the following corollary.

**Corollary A.5** (Implicit Function Theorem for optimization subroutines)
*Let $u : \mathbb{R}^n \times \mathbb{R}^m \to \mathbb{R}$ be a twice continuously differentiable function such that the choice $h = \nabla_y u$ satisfies the hypotheses of Proposition A.4 at some point $(\bar{x}, \bar{y})$, and define $y^*$ as in Proposition A.4. Then we have*

$$\nabla y^*(x) = -\nabla^2_{xy} u\left(x, y^*(x)\right) \left(\nabla^2_{yy} u\left(x, y^*(x)\right)\right)^{-1}, \qquad \forall\, x \in S_{\bar{x}}.$$

# B    Exponential families

In this section we set up notation for exponential families and outline some basic results. Throughout this section we take all densities to be absolutely continuous with respect to the appropriate Lebesgue measure (when the underlying set $\mathcal{X}$ is Euclidean space) or counting measure (when $\mathcal{X}$ is discrete), and denote the Borel $\sigma$-algebra of a set $\mathcal{X}$ as $\mathcal{B}(\mathcal{X})$ (generated by Euclidean and discrete topologies, respectively). We assume measurability of all functions as necessary.

Given a statistic function $t_x : \mathcal{X} \to \mathbb{R}^n$ and a base measure $\nu_{\mathcal{X}}$, we can define an exponential family of probability densities on $\mathcal{X}$ relative to $\nu_{\mathcal{X}}$ and indexed by natural parameter $\eta_x \in \mathbb{R}^n$ by

$$p(x \,|\, \eta_x) \propto \exp\left\{\langle \eta_x,\, t_x(x)\rangle\right\}, \quad \forall \eta_x \in \mathbb{R}^n,$$

where $\langle \cdot,\, \cdot \rangle$ is the standard inner product on $\mathbb{R}^n$. We also define the partition function as

$$Z_x(\eta_x) \triangleq \int \exp\left\{\langle \eta_x,\, t_x(x)\rangle\right\} \nu_{\mathcal{X}}(dx)$$

and define $H \subseteq \mathbb{R}^n$ to be the set of all normalizable natural parameters,

$$H \triangleq \left\{\eta \in \mathbb{R}^n : Z_x(\eta) < \infty\right\}.$$

We can write the normalized probability density as

$$p(x \,|\, \eta) = \exp\left\{\langle \eta_x,\, t_x(x)\rangle - \log Z_x(\eta_x)\right\}. \tag{2}$$

We say that an exponential family is *regular* if $H$ is open, and *minimal* if there is no $\eta \in \mathbb{R}^n \setminus \{0\}$ such that $\langle \eta,\, t_x(x)\rangle = 0$ ($\nu_{\mathcal{X}}$-a.e.). We assume all families are regular and minimal.[2] Finally, when we parameterize the family with some other coordinates $\theta$, we write the natural parameter as a continuous function $\eta_x(\theta)$ and write the density as

$$p(x \,|\, \theta) = \exp\left\{\langle \eta_x(\theta),\, t_x(x)\rangle - \log Z_x(\eta_x(\theta))\right\}$$

and take $\Theta = \eta_x^{-1}(H)$ to be the open set of parameters that correspond to normalizable densities. We summarize this notation in the following definition.

**Definition B.1** (Exponential family of densities)
*Given a measure space $(\mathcal{X}, \mathcal{B}(\mathcal{X}), \nu_{\mathcal{X}})$, a statistic function $t_x : \mathcal{X} \to \mathbb{R}^n$, and a natural parameter function $\eta_x : \Theta \to \mathbb{R}^n$, the corresponding* exponential family of densities *relative to $\nu_{\mathcal{X}}$ is*

$$p(x \,|\, \theta) = \exp\left\{\langle \eta_x(\theta),\, t_x(x)\rangle - \log Z_x(\eta_x(\theta))\right\},$$

*where*

$$\log Z_x(\eta_x) \triangleq \log \int \exp\left\{\langle \eta_x,\, t_x(x)\rangle\right\} \nu_{\mathcal{X}}(dx)$$

*is the log partition function.*

When we write exponential families of densities for different random variables, we change the subscripts on the statistic function, natural parameter function, and log partition function to correspond to the symbol used for the random variable. When the corresponding random variable is clear from context, we drop the subscripts to simplify notation.

The next proposition shows that the log partition function of an exponential family generates cumulants of the statistic.

**Proposition B.2** (Gradients of $\log Z$ and expected statistics)
*The gradient of the log partition function of an exponential family gives the expected sufficient statistic,*

$$\nabla \log Z(\eta) = \mathbb{E}_{p(x \mid \eta)}\left[t(x)\right],$$

*where the expectation is over the random variable $x$ with density $p(x \mid \eta)$. More generally, the moment generating function of $t(x)$ can be written*

$$\mathrm{M}_{t(x)}(s) \triangleq \mathbb{E}_{p(x \mid \eta)}\left[e^{\langle s, t(x)\rangle}\right] = e^{\log Z(\eta + s) - \log Z(\eta)}$$

*and so derivatives of $\log Z$ give cumulants of $t(x)$, where the first cumulant is the mean and the second and third cumulants are the second and third central moments, respectively.*

Given an exponential family of densities on $\mathcal{X}$ as in Definition B.1, we can define a related exponential family of densities on $\Theta$ by defining a statistic function $t_\theta(\theta)$ in terms of the functions $\eta_x(\theta)$ and $\log Z_x(\eta_x(\theta))$.

**Definition B.3** (Natural exponential family conjugate prior)
*Given the exponential family $p(x \mid \theta)$ of Definition B.1, define the statistic function $t_\theta : \Theta \to \mathbb{R}^{n+1}$ as the concatenation*

$$t_\theta(\theta) \triangleq (\eta_x(\theta), -\log Z_x(\eta_x(\theta))),$$

*where the first $n$ coordinates of $t_\theta(\theta)$ are given by $\eta_x(\theta)$ and the last coordinate is given by $-\log Z_x(\eta_x(\theta))$. We call the exponential family with statistic $t_\theta(\theta)$ the natural exponential family conjugate prior to the density $p(x \mid \theta)$ and write*

$$p(\theta) = \exp\left\{\langle \eta_\theta, t_\theta(\theta)\rangle - \log Z_\theta(\eta_\theta)\right\}$$

*where $\eta_\theta \in \mathbb{R}^{n+1}$ and the density is taken relative to some measure $\nu_\Theta$ on $(\Theta, \mathcal{B}(\Theta))$.*

Notice that using $t_\theta(\theta)$ we can rewrite the original density $p(x \mid \theta)$ as

$$\begin{aligned} p(x \mid \theta) &= \exp\left\{\langle \eta_x(\theta), t_x(x)\rangle - \log Z_x(\eta_x(\theta))\right\} \\ &= \exp\left\{\langle t_\theta(\theta), (t_x(x), 1)\rangle\right\}. \end{aligned}$$

This relationship is useful in Bayesian inference: when the exponential family $p(x \mid \theta)$ is a likelihood function and the family $p(\theta)$ is used as a prior, the pair enjoy a convenient conjugacy property, as summarized in the next proposition.

**Proposition B.4** (Conjugacy)
*Let the densities $p(x \mid \theta)$ and $p(\theta)$ be defined as in Definitions B.1 and B.3, respectively. We have the relations*

$$\begin{aligned} p(\theta, x) &= \exp\left\{\langle \eta_\theta + (t_x(x), 1),\, t_\theta(\theta)\rangle - \log Z_\theta(\eta_\theta)\right\} & (3) \\ p(\theta \mid x) &= \exp\left\{\langle \eta_\theta + (t_x(x), 1),\, t_\theta(\theta)\rangle - \log Z_\theta(\eta_\theta + (t_x(x), 1))\right\} \end{aligned}$$

*and hence in particular the posterior $p(\theta \mid x)$ is in the same exponential family as $p(\theta)$ with the natural parameter $\eta_\theta + (t_x(x), 1)$. Similarly, with multiple likelihood terms $p(x_i \mid \theta)$ for $i = 1, 2, \ldots, N$ we have*

$$p(\theta)\prod_{i=1}^{N} p(x_i \mid \theta) = \exp\left\{\langle \eta_\theta + \sum_{i=1}^{N}(t_x(x_i), 1),\, t_\theta(\theta)\rangle - \log Z_\theta(\eta_\theta)\right\}. \qquad (4)$$

Finally, we give a few more exponential family properties that are useful for gradient-based optimization algorithms and variational inference. In particular, we note that the Fisher information matrix of an exponential family can be computed as the Hessian matrix of its log partition function, and that the KL divergence between two members of the same exponential family has a simple expression.

**Definition B.5** (Score vector and Fisher information matrix)
*Given a family of densities $p(x \mid \theta)$ indexed by a parameter $\theta$, the* score *vector $v(x, \theta)$ is the gradient of the log density with respect to the parameter,*

$$v(x, \theta) \triangleq \nabla_\theta \log p(x \mid \theta),$$

*and the* Fisher information matrix *for the parameter $\theta$ is the covariance of the score,*

$$I(\theta) \triangleq \mathbb{E}\left[ v(x, \theta) v(x, \theta)^{\mathsf{T}} \right],$$

*where the expectation is taken over the random variable $x$ with density $p(x \mid \theta)$, and where we have used the identity $\mathbb{E}[v(x, \theta)] = 0$.*

**Proposition B.6** (Score and Fisher information for exponential families)
*Given an exponential family of densities $p(x \mid \eta)$ indexed by the natural parameter $\eta$, as in Eq. (2), the score with respect to the natural parameter is given by*

$$v(x, \eta) = \nabla_\eta \log p(x \mid \eta) = t(x) - \nabla \log Z(\eta)$$

*and the Fisher information matrix is given by*

$$I(\eta) = \nabla^2 \log Z(\eta).$$

**Proposition B.7** (KL divergence in an exponential family)
*Given an exponential family of densities $p(x \mid \eta)$ indexed by the natural parameter $\eta$, as in Eq. (2), and two particular members with natural parameters $\eta_1$ and $\eta_2$, respectively, the KL divergence from one to the other is*

$$\mathrm{KL}(p(x \mid \eta_1) \parallel p(x \mid \eta_2)) \triangleq \mathbb{E}_{p(x \mid \eta_1)} \left[ \log \frac{p(x \mid \eta_1)}{p(x \mid \eta_2)} \right] \tag{5}$$

$$= \langle \eta_1 - \eta_2, \ \nabla \log Z(\eta_1) \rangle - (\log Z(\eta_1) - \log Z(\eta_2)).$$

## C  Natural gradient SVI for exponential families

In this section we give a derivation of the natural gradient stochastic variational inference (SVI) method of Hoffman et al. [5] using our notation. We extend the algorithm in Section D.

### C.1  SVI objective

Let $p(x, y \mid \theta)$ be an exponential family and $p(\theta)$ be its corresponding natural exponential family prior as in Definitions B.1 and B.3, writing

$$p(\theta) = \exp \left\{ \langle \eta_\theta^0, \ t_\theta(\theta) \rangle - \log Z_\theta(\eta_\theta^0) \right\}$$

$$p(x, y \mid \theta) = \exp \left\{ \langle \eta_{xy}^0(\theta), \ t_{xy}(x, y) \rangle - \log Z_{xy}(\eta_{xy}^0(\theta)) \right\}$$

$$= \exp \left\{ \langle t_\theta(\theta), \ (t_{xy}(x, y), 1) \rangle \right\} \tag{6}$$

where we have used $t_\theta(\theta) = \left( \eta_{xy}^0(\theta), -\log Z_{xy}(\eta_{xy}^0(\theta)) \right)$ in Eq. (6).

Given a fixed observation $y$, for any density $q(\theta, x) = q(\theta)q(x)$ we have

$$\log p(y) = \mathbb{E}_{q(\theta)q(x)} \left[ \log \frac{p(\theta)p(x, y \mid \theta)}{q(\theta)q(x)} \right] + \mathrm{KL}(q(\theta)q(x) \parallel p(\theta, x \mid y))$$

$$\geq \mathbb{E}_{q(\theta)q(x)} \left[ \log \frac{p(\theta)p(x, y \mid \theta)}{q(\theta)q(x)} \right]$$

where we have used the fact that the KL divergence is always nonnegative. Therefore to choose $q(\theta)q(x)$ to minimize the KL divergence to the posterior $p(\theta, x \mid y)$ we define the mean field variational inference objective as

$$\mathcal{L}\left[ q(\theta)q(x) \right] \triangleq \mathbb{E}_{q(\theta)q(x)} \left[ \log \frac{p(\theta)p(x, y \mid \theta)}{q(\theta)q(x)} \right] \tag{7}$$

and the mean field variational inference problem as

$$\max {}_{\cdot q(\theta)q(x)} \mathcal{L}\left[ q(\theta)q(x) \right]. \tag{8}$$

The following proposition shows that because of the exponential family conjugacy structure, we can fix the parameterization of $q(\theta)$ and still optimize over all possible densities without loss of generality.

**Proposition C.1** (Optimal form of the global variational factor)
*Given the mean field optimization problem Eq. ([8](#)), for any fixed $q(x)$ the optimal factor $q(\theta)$ is detetermined ($\nu_{\Theta}$-a.e.) by*

$$q(\theta) \propto \exp\left\{\left\langle \eta_{\theta}^0 + \mathbb{E}_{q(x)}\left[\,(t_{xy}(x,y),1)\,\right], t_{\theta}(\theta) \right\rangle\right\}.$$

*In particular, the optimal $q(\theta)$ is in the same exponential family as the prior $p(\theta)$.*

This proposition follows immediately from a more general lemma, which we reuse in the sequel.

**Lemma C.2** (Optimizing a mean field factor)
*Let $p(a,b,c)$ be a joint density and let $q(a)$, $q(b)$, and $q(c)$ be mean field factors. Consider the mean field variational inference objective*

$$\mathbb{E}_{q(a)q(b)q(c)}\left[\log \frac{p(a,b,c)}{q(a)q(b)q(c)}\right].$$

*For fixed $q(a)$ and $q(c)$, the partially optimal factor $q^*(b)$ over all possible densities,*

$$q^*(b) \triangleq \underset{q(b)}{\arg\max}\, \mathbb{E}_{q(a)q(b)q(c)}\left[\log \frac{p(a,b,c)}{q(a)q(b)q(c)}\right], \tag{9}$$

*is defined (almost everywhere) by*

$$q^*(b) \propto \exp\left\{\mathbb{E}_{q(a)q(c)}\log p(a,b,c)\right\}.$$

*In particular, if $p(c\,|\,b,a)$ is an exponential family with $p(b\,|\,a)$ its natural exponential family conjugate prior, and $\log p(b,c\,|\,a)$ is a multilinear polynomial in the statistics $t_b(b)$ and $t_c(c)$, written*

$$p(b\,|\,a) = \exp\left\{\langle \eta_b^0(a),\, t_b(b)\rangle - \log Z_b(\eta_b^0(a))\right\},$$
$$p(c\,|\,b,a) = \exp\left\{\langle \eta_c^0(b,a),\, t_c(c)\rangle - \log Z_c(\eta_c^0(b,a))\right\}$$
$$= \exp\left\{\langle t_b(b), \eta_c^0(a)^{\mathsf{T}}(t_c(c),1)\rangle\right\},$$

*for some matrix $\eta_c^0(a)$, then the optimal factor can be written*

$$q^*(b) = \exp\left\{\langle \eta_b^*,\, t_b(b)\rangle - \log Z_b(\eta_b^*)\right\}, \qquad \eta_b^* \triangleq \mathbb{E}_{q(a)}\eta_b^0(a) + \mathbb{E}_{q(a)q(c)}\eta_c^0(a)^{\mathsf{T}}(t_c(c),1).$$

*As a special case, when $c$ is conditionally independent of $b$ given $a$, so that $p(c\,|\,b,a) = p(c\,|\,b)$, then*

$$p(c\,|\,b) = \exp\left\{\langle t_b(b), (t_c(c),1)\rangle\right\}, \qquad \eta_b^* \triangleq \mathbb{E}_{q(a)}\eta_b^0(a) + \mathbb{E}_{q(c)}(t_c(c),1).$$

*Proof.* Rewrite the objective in Eq. ([9](#)), dropping terms that are constant with respect to $q(b)$, as

$$\mathbb{E}_{q(a)q(b)q(c)}\left[\log \frac{p(a,b,c)}{q(b)}\right] = \mathbb{E}_{q(b)}\left[\mathbb{E}_{q(a)q(c)}\log p(a,b,c) - \log q(c)\right]$$
$$= \mathbb{E}_{q(b)}\left[\log \exp \mathbb{E}_{q(a)q(c)}\log p(a,b,c) - \log q(c)\right]$$
$$= -\mathbb{E}_{q(b)}\left[\frac{q(b)}{\widetilde{p}(b)}\right] + \text{const}$$
$$= -\,\mathrm{KL}(q(b)\,\|\,\widetilde{p}(b)) + \text{const},$$

where we have defined a new density $\widetilde{p}(b) \propto \exp\left\{\mathbb{E}_{q(a)q(c)}\log p(a,b,c)\right\}$. We can maximize the objective by setting the KL divergence to zero, choosing $q(b) \propto \exp\left\{\mathbb{E}_{q(a)q(c)}\log p(a,b,c)\right\}$. The rest follows from plugging in the exponential family densities. $\square$

Proposition [C.1](#) justifies parameterizing the density $q(\theta)$ with variational natural parameters $\eta_{\theta}$ as

$$q(\theta) = \exp\left\{\langle \eta_{\theta},\, t_{\theta}(\theta)\rangle - \log Z_{\theta}(\eta_{\theta})\right\}$$

where the statistic function $t_{\theta}$ and the log partition function $\log Z_{\theta}$ are the same as in the prior family $p(\theta)$. Using this parameterization, we can define the mean field objective as a function of the parameters $\eta_{\theta}$, partially optimizing over $q(x)$,

$$\mathcal{L}(\eta_{\theta}) \triangleq \max_{q(x)} \mathbb{E}_{q(\theta)q(x)}\left[\log \frac{p(\theta)p(x,y\,|\,\theta)}{q(\theta)q(x)}\right]. \tag{10}$$

The partial optimization over $q(x)$ in Eq. ([10](#)) should be read as choosing $q(x)$ to be a local partial optimizer of Eq. ([7](#)); in general, it may be intractable to find a global partial optimizer, and the results that follow use only first-order stationary conditions on $q(x)$. We refer to this objective function, where we locally partially optimize the mean field objective Eq. ([7](#)) over $q(x)$, as the SVI objective.

## C.2  Easy natural gradients of the SVI objective

By again leveraging the conjugate exponential family structure, we can write a simple expression for the gradient of the SVI objective, and even for its natural gradient.

**Proposition C.3** (Gradient of the SVI objective)
*Let the SVI objective $\mathcal{L}(\eta_\theta)$ be defined as in Eq. (10). Then the gradient $\nabla \mathcal{L}(\eta_\theta)$ is*

$$\nabla \mathcal{L}(\eta_\theta) = \left( \nabla^2 \log Z_\theta(\eta_\theta) \right) \left( \eta_\theta^0 + \mathbb{E}_{q^*(x)} \left[ (t_{xy}(x,y), 1) \right] - \eta_\theta \right)$$

*where $q^*(x)$ is a local partial optimizer of the mean field objective Eq. (7) for fixed global variational parameters $\eta_\theta$.*

*Proof.* First, note that because $q^*(x)$ is a local partial optimizer for Eq. (7) by Proposition A.3, we have

$$\nabla \mathcal{L}(\eta_\theta) = \nabla_{\eta_\theta} \mathbb{E}_{q(\theta)q^*(x)} \left[ \log \frac{p(\theta)p(x,y \mid \theta)}{q(\theta)q^*(x)} \right].$$

Next, we use the conjugate exponential family structure and Proposition B.4, Eq. (3), to expand

$$\mathbb{E}_{q(\theta)q^*(x)} \left[ \log \frac{p(\theta)p(x,y \mid \theta)}{q(\theta)q^*(x)} \right] = \langle \eta_\theta^0 + \mathbb{E}_{q^*(x)}(t_{xy}(x,y), 1) - \eta_\theta, \ \mathbb{E}_{q(\theta)}[t_\theta(\theta)] \rangle$$
$$- \left( \log Z_\theta(\eta_\theta^0) - \log Z_\theta(\eta_\theta) \right).$$

Note that we can use Proposition B.2 to replace $\mathbb{E}_{q(\theta)}[t_\theta(\theta)]$ with $\nabla \log Z_\theta(\eta_\theta)$. Differentiating with respect to $\eta_\theta$ and using the product rule, we have

$$\nabla \mathcal{L}(\eta_\theta) = \nabla^2 \log Z_\theta(\eta_\theta) \left( \eta_\theta^0 + \mathbb{E}_{q^*(x)}(t_{xy}(x,y), 1) - \eta_\theta \right)$$
$$- \nabla \log Z_\theta(\eta_\theta) + \nabla \log Z_\theta(\eta_\theta)$$
$$= \nabla^2 \log Z_\theta(\eta_\theta) \left( \eta_\theta^0 + \mathbb{E}_{q^*(x)}(t_{xy}(x,y), 1) - \eta_\theta \right).$$

$\square$

As an immediate result of Proposition C.3, the natural gradient [6] defined by

$$\widetilde{\nabla} \mathcal{L}(\eta_\theta) \triangleq \left( \nabla^2 \log Z_\theta(\eta_\theta) \right)^{-1} \nabla \mathcal{L}(\eta_\theta)$$

has an even simpler expression.

**Corollary C.4** (Natural gradient of the SVI objective)
*The natural gradient of the SVI objective Eq. (10) is*

$$\widetilde{\nabla} \mathcal{L}(\eta_\theta) = \eta_\theta^0 + \mathbb{E}_{q^*(x)} \left[ (t_{xy}(x,y), 1) \right] - \eta_\theta.$$

The natural gradient corrects for a kind of curvature in the variational family and is invariant to reparameterization of the family [7]. As a result, natural gradient ascent is effectively a second-order quasi-Newton optimization algorithm, and using natural gradientds can greatly accelerate the convergence of gradient-based optimization algorithms [8, 9]. It is a remarkable consequence of the exponential family structure that natural gradients of the partially optimized mean field objective with respect to the global variational parameters can be computed efficiently (without any backward pass as would be required in generic reverse-mode differentiation). Indeed, the exponential family conjugacy structure makes the natural gradient of the SVI objective even easier to compute than the flat gradient.

## C.3  Stochastic natural gradients for large datasets

The real utility of natural gradient SVI is in its application to large datasets. Consider the model composed of global latent variables $\theta$, local latent variables $x = \{x_n\}_{n=1}^N$, and data $y = \{y_n\}_{n=1}^N$,

$$p(\theta, x, y) = p(\theta) \prod_{n=1}^N p(x_n, y_n \mid \theta),$$

where each $p(x_n, y_n \,|\, \theta)$ is a copy of the same likelihood function with conjugate prior $p(\theta)$. For fixed observations $y = \{y_n\}_{n=1}^N$, let

$$q(\theta, x) = q(\theta) \prod_{n=1}^N q(x_n)$$

be a variational family to approximate the posterior $p(\theta, x \,|\, y)$ and consider the SVI objective given by Eq. (10). Using Eq. (4) of Proposition B.4, it is straightforward to extend the natural gradient expression in Corollary C.4 to an unbiased Monte Carlo estimate which samples terms in the sum over data points.

**Corollary C.5** (Unbiased Monte Carlo estimate of the SVI natural gradient)
*Using the model and variational family*

$$p(\theta, x, y) = p(\theta) \prod_{n=1}^N p(x_n, y_n \,|\, \theta), \qquad q(\theta)q(x) = q(\theta) \prod_{n=1}^N q(x_n),$$

*where $p(\theta)$ and $p(x_n, y_n \,|\, \theta)$ are a conjugate pair of exponential families, define $\mathcal{L}(\eta_\theta)$ as in Eq. (10). Let the random index $\hat{n}$ be sampled from the set $\{1, 2, \ldots, N\}$ and let $p_n > 0$ be the probability it takes value $n$. Then*

$$\widetilde{\nabla}\mathcal{L}(\eta_\theta) = \mathbb{E}_{\hat{n}} \left[ \eta_\theta^0 + \frac{1}{p_{\hat{n}}} \mathbb{E}_{q^*(x_{\hat{n}})}[\, (t_{xy}(x_{\hat{n}}, y_{\hat{n}}), 1) \,] - \eta_\theta \right],$$

*where $q^*(x_{\hat{n}})$ is a local partial optimizer of $\mathcal{L}$ given $q(\theta)$.*

*Proof.* Taking expectation over the index $\hat{n}$, we have

$$\mathbb{E}_{\hat{n}} \left[ \frac{1}{p_{\hat{n}}} \mathbb{E}_{q^*(x_{\hat{n}})}[\, (t_{xy}(x_{\hat{n}}, y_{\hat{n}}), 1) \,] \right] = \sum_{n=1}^N \frac{p_n}{p_n} \mathbb{E}_{q^*(x_n)}[\, (t_{xy}(x_n, y_n), 1) \,]$$

$$= \sum_{n=1}^N \mathbb{E}_{q^*(x_n)}[\, (t_{xy}(x_n, y_n), 1) \,].$$

The remainder of the proof follows from Proposition B.4 and the same argument as in Proposition C.3. □

The unbiased stochastic gradient developed in Corollary C.5 can be used in a scalable stochastic gradient ascent algorithm. To simplify notation, in the following sections we drop the notation for multiple likelihood terms $p(x_n, y_n \,|\, \theta)$ for $n = 1, 2, \ldots, N$ and return to working with a single likelihood term $p(x, y \,|\, \theta)$. The extension to multiple likelihood terms is immediate.

## C.4 Conditinally conjugate models and block updating

The model classes often considered for natural gradient SVI, and the main model classes we consider here, have additional conjugacy structure in the local latent variables. In this section we introduce notation for this extra structure in terms of the additional local latent variables $z$ and discuss the local block coordinate optimization that is often performed to compute the factor $q^*(z)q^*(x)$ for use in the natural gradient expression.

Let $p(z, x, y \,|\, \theta)$ be an exponential family and $p(\theta)$ be its corresponding natural exponential family conjugate prior, writing

$$p(\theta) = \exp\left\{ \langle \eta_\theta^0, t_\theta(\theta) \rangle - \log Z_\theta(\eta_\theta^0) \right\}, \tag{11}$$

$$p(z, x, y \,|\, \theta) = \exp\left\{ \langle \eta_{zxy}^0(\theta), t_{zxy}(z, x, y) \rangle - \log Z_{zxy}(\eta_{zxy}^0(\theta)) \right\}$$

$$= \exp\left\{ \langle t_\theta(\theta), (t_{zxy}(z, x, y), 1) \rangle \right\}, \tag{12}$$

where we have used $t_\theta(\theta) = \left( \eta_{zxy}^0(\theta), -\log Z_{zxy}(\eta_{zxy}^0(\theta)) \right)$ in Eq. (12). Additionally, let $t_{zxy}(z, x, y)$ be a multilinear polynomial in the statistics functions $t_x(x)$, $t_y(y)$, and $t_z(z)$, let $p(z \,|\, \theta)$,

$p(x \mid z, \theta)$, and $p(y \mid x, z, \theta) = p(y \mid x, \theta)$ be exponential families, and let $p(z \mid \theta)$ be a conjugate prior to $p(x \mid z, \theta)$ and $p(x \mid z, \theta)$ be a conjugate prior to $p(y \mid x, \theta)$, so that

$$p(z \mid \theta) = \exp \left\{ \langle \eta_z^0(\theta),\, t_z(z) \rangle - \log Z_z(\eta_z^0(\theta)) \right\}, \tag{13}$$

$$p(x \mid z, \theta) = \exp \left\{ \langle \eta_x^0(z, \theta),\, t_x(x) \rangle - \log Z_x(\eta_x^0(z, \theta)) \right\}$$
$$= \exp \left\{ \langle t_z(z),\, \eta_x^0(\theta)^\mathsf{T}(t_x(x), 1) \rangle \right\}, \tag{14}$$

$$p(y \mid x, \theta) = \exp \left\{ \langle \eta_y^0(x, \theta),\, t_y(y) \rangle - \log Z_y(\eta_y^0(x, z, \theta)) \right\}$$
$$= \exp \left\{ \langle t_x(x),\, \eta_y^0(\theta)^\mathsf{T}(t_y(y), 1) \rangle \right\}, \tag{15}$$

for some matrices $\eta_x^0(\theta)$ and $\eta_y^0(\theta)$.

This model class includes many common models, including the latent Dirichlet allocation, switching linear dynamical systems with linear-Gaussian emissions, and mixture models and hidden Markov models with exponential family emissions. The conditionally conjugate structure is both powerful and restrictive: while it potentially limits the expressiveness of the model class, it enables block coordinate optimization with very simple and fast updates, as we show next. When conditionally conjugate structure is not present, these local optimizations can instead be performed with generic gradient-based methods and automatic differentiation [10].

**Proposition C.6** (Unconstrained block coordinate ascent on $q(z)$ and $q(x)$)
*Let $p(\theta, z, x, y)$ be a model as in Eqs. (11)-(15), and for fixed data $y$ let $q(\theta)q(z)q(x)$ be a corresponding mean field variational family for approximating the posterior $p(\theta, z, x \mid y)$, with*

$$q(\theta) = \exp \left\{ \langle \eta_\theta,\, t_\theta(\theta) \rangle - \log Z_\theta(\eta_\theta) \right\},$$
$$q(z) = \exp \left\{ \langle \eta_z,\, t_z(z) \rangle - \log Z_z(\eta_z) \right\},$$
$$q(x) = \exp \left\{ \langle \eta_x,\, t_x(x) \rangle - \log Z_x(\eta_x) \right\},$$

*and with the mean field variational inference objective*

$$\mathcal{L}[\, q(\theta)q(z)q(x)\,] = \mathbb{E}_{q(\theta)q(z)q(x)} \left[ \log \frac{p(\theta)p(z \mid \theta)p(x \mid z, \theta)p(y \mid x, z, \theta)}{q(\theta)q(z)q(x)} \right].$$

*Fixing the other factors, the partial optimizers $q^*(z)$ and $q^*(x)$ for $\mathcal{L}$ over all possible densities are given by*

$$q^*(z) \triangleq \underset{q(z)}{\arg\max}\, \mathcal{L}[\, q(\theta)q(z)q(x)\,] = \exp \left\{ \langle \eta_z^*,\, t_z(z) \rangle - \log Z_z(\eta_z^*) \right\},$$
$$q^*(x) \triangleq \underset{q(x)}{\arg\max}\, \mathcal{L}[\, q(\theta)q(z)q(x)\,] = \exp \left\{ \langle \eta_x^*,\, t_x(x) \rangle - \log Z_x(\eta_x^*) \right\},$$

*with*

$$\eta_z^* = \mathbb{E}_{q(\theta)} \eta_z^0(\theta) + \mathbb{E}_{q(\theta)q(x)} \eta_x^0(\theta)^\mathsf{T}(t_x(x), 1), \tag{16}$$

$$\eta_x^* = \mathbb{E}_{q(\theta)q(z)} \eta_x^0(\theta) t_z(z) + \mathbb{E}_{q(\theta)} \eta_y^0(\theta)^\mathsf{T}(t_y(y), 1). \tag{17}$$

*Proof.* This proposition is a consequence of Lemma C.2 and the conjugacy structure. □

Proposition C.6 gives an efficient block coordinate ascent algorithm: for fixed $\eta_\theta$, by alternatively updating $\eta_z$ and $\eta_x$ according to Eqs. (16)-(17) we are guaranteed to converge to a stationary point that is partially optimal in the parameters of each factor. In addition, performing each update requires only computing expected sufficient statistics in the variational factors, which means evaluating $\nabla \log Z_\theta(\eta_\theta)$, $\nabla \log Z_z(\eta_z)$, and $\nabla \log Z_x(\eta_x)$, quantities that be computed anyway in a gradient-based optimization routine. The block coordinate ascent procedure leveraging this conditional conjugacy structure is thus not only efficient but also does not require a choice of step size.

Note in particular that this procedure produces parameters $\eta_z^*(\eta_\theta)$ and $\eta_x^*(\eta_\theta)$ that are partially optimal (and hence stationary) for the objective. That is, defining the parameterized mean field variational inference objective as $L(\eta_\theta, \eta_z, \eta_x) = \mathcal{L}[\, q(\theta)q(z)q(x)\,]$, for fixed $\eta_\theta$ the block coordinate ascent procedure has limit points $\eta_z^*$ and $\eta_x^*$ that satisfy

$$\nabla_{\eta_z} \mathcal{L}(\eta_\theta, \eta_z^*(\eta_\theta), \eta_x^*(\eta_\theta)) = 0, \qquad\qquad \nabla_{\eta_x} \mathcal{L}(\eta_\theta, \eta_z^*(\eta_\theta), \eta_x^*(\eta_\theta)) = 0.$$

# D The SVAE objective and its gradients

In this section we define the SVAE variational lower bound and show how to efficiently compute unbiased stochastic estimates of its gradients, including an unbiased estimate of the natural gradient with respect to the variational parameters with conjugacy structure. The setup here parallels the setup for natural gradient SVI in Section C, but while SVI is restricted to complete-data conjugate models, here we consider more general likelihood models.

## D.1 SVAE objective

Let $p(x \mid \theta)$ be an exponential family and let $p(\theta)$ be its corresponding natural exponential family conjugate prior, as in Definitions B.1 and B.3, writing

$$p(\theta) = \exp\left\{\langle \eta_\theta^0, t_\theta(\theta)\rangle - \log Z_\theta(\eta_\theta^0)\right\}, \tag{18}$$

$$p(x \mid \theta) = \exp\left\{\langle \eta_x^0(\theta), t_x(x)\rangle - \log Z_x(\eta_x^0(\theta))\right\}$$
$$= \exp\left\{\langle t_\theta(\theta), (t_x(x), 1)\rangle\right\}, \tag{19}$$

where we have used $t_\theta(\theta) = \left(\eta_x^0(\theta), -\log Z_x(\eta_x^0(\theta))\right)$ in Eq. (19). Let $p(y \mid x, \gamma)$ be a general family of densities (not necessarily an exponential family) and let $p(\gamma)$ be an exponential family prior on its parameters of the form

$$p(\gamma) = \exp\left\{\langle \eta_\gamma^0, t_\gamma(\gamma)\rangle - \log Z_\gamma(\eta_\gamma^0)\right\}.$$

For fixed $y$, consider the mean field family of densities $q(\theta, \gamma, x) = q(\theta)q(\gamma)q(x)$ and the mean field variational inference objective

$$\mathcal{L}[\, q(\theta)q(\gamma)q(x)\,] \triangleq \mathbb{E}_{q(\theta)q(\gamma)q(x)}\left[\log \frac{p(\theta)p(\gamma)p(x \mid \theta)p(y \mid x, \gamma)}{q(\theta)q(\gamma)q(x)}\right]. \tag{20}$$

By the same argument as in Proposition C.1, without loss of generality we can take the global factor $q(\theta)$ to be in the same exponential family as the prior $p(\theta)$, and we denote its natural parameters by $\eta_\theta$, writing

$$q(\theta) = \exp\left\{\langle \eta_\theta, t_\theta(\theta)\rangle - \log Z_\theta(\eta_\theta)\right\}.$$

We restrict $q(\gamma)$ to be in the same exponential family as $p(\gamma)$ with natural parameters $\eta_\gamma$, writing

$$q(\gamma) = \exp\left\{\langle \eta_\gamma, t_\gamma(\gamma)\rangle - \log Z_\gamma(\eta_\gamma)\right\}.$$

Finally, we restrict[3] $q(x)$ to be in the same exponential family as $p(x \mid \theta)$, writing its natural parameter as $\eta_x$. Using these explicit variational natural parameters, we rewrite the mean field variational inference objective in Eq. (20) as

$$\mathcal{L}(\eta_\theta, \eta_\gamma, \eta_x) \triangleq \mathbb{E}_{q(\theta)q(\gamma)q(x)}\left[\log \frac{p(\theta)p(\gamma)p(x \mid \theta)p(y \mid x, \gamma)}{q(\theta)q(\gamma)q(x)}\right]. \tag{21}$$

To perform efficient optimization in the objective $\mathcal{L}$ defined in Eq. (21), we consider choosing the variational parameter $\eta_x$ as a function of the other parameters $\eta_\theta$ and $\eta_\gamma$. One natural choice is to set $\eta_x$ to be a local partial optimizer of $\mathcal{L}$, as in Section C. However, finding a local partial optimizer may be computationally expensive for general densities $p(y \mid x, \gamma)$, and in the large data setting this expensive optimization would have to be performed for each stochastic gradient update. Instead, we choose $\eta_x$ by optimizing over a surrogate objective $\widehat{\mathcal{L}}$, which we design using exponential family structure to be both easy to optimize and to share curvature properties with the mean field objective $\mathcal{L}$. The surrogate objective $\widehat{\mathcal{L}}$ is

$$\widehat{\mathcal{L}}(\eta_\theta, \eta_\gamma, \eta_x, \phi) \triangleq \mathbb{E}_{q(\theta)q(\gamma)q(x)}\left[\log \frac{p(\theta)p(\gamma)p(x \mid \theta)\exp\{\psi(x; y, \phi)\}}{q(\theta)q(\gamma)q(x)}\right]$$
$$= \mathbb{E}_{q(\theta)q(x)}\left[\log \frac{p(\theta)p(x \mid \theta)\exp\{\psi(x; y, \phi)\}}{q(\theta)q(x)}\right] + \text{const}, \tag{22}$$

where the constant does not depend on $\eta_x$. We define the function $\psi(x; y, \phi)$ to have a form related to the exponential family $p(x \mid \theta)$,

$$\psi(x; y, \phi) \triangleq \langle r(y; \phi), \ t_x(x) \rangle, \tag{23}$$

where $\{r(y; \phi)\}_{\phi \in \mathbb{R}^m}$ is some class of functions parameterized by $\phi \in \mathbb{R}^m$, which we assume only to be continuously differentiable in $\phi$. We call $r(y; \phi)$ the *recognition model*. We define $\eta_x^*(\eta_\theta, \phi)$ to be a local partial optimizer of $\widehat{\mathcal{L}}$,

$$\eta_x^*(\eta_\theta, \phi) \triangleq \arg\min_{\eta_x} \widehat{\mathcal{L}}(\eta_\theta, \eta_\gamma, \eta_x, \phi),$$

where the notation above should be interpreted as choosing $\eta_x^*(\eta_\theta, \phi)$ to be a local argument of maximum. The results to follow rely only on necessary first-order conditions for unconstrained local optimality.

Given this choice of function $\eta_x^*(\eta_\theta, \phi)$, we define the SVAE objective to be

$$\mathcal{L}_{\text{SVAE}}(\eta_\theta, \eta_\gamma, \phi) \triangleq \mathcal{L}(\eta_\theta, \eta_\gamma, \eta_x^*(\eta_\theta, \phi)), \tag{24}$$

where $\mathcal{L}$ is the mean field variational inference defined in Eq. (21), and we define the SVAE optimization problem to be

$$\max{}_{\eta_\theta, \eta_\gamma, \phi} \mathcal{L}_{\text{SVAE}}(\eta_\theta, \eta_\gamma, \phi).$$

We summarize these definitions in the following.

**Definition D.1** (SVAE objective)
*Let $\mathcal{L}$ denote the mean field variational inference objective*

$$\mathcal{L}[\, q(\theta)q(\gamma)q(x) \,] \triangleq \mathbb{E}_{q(\theta)q(\gamma)q(x)} \left[ \log \frac{p(\theta)p(\gamma)p(x \mid \theta)p(y \mid x, \gamma)}{q(\theta)q(\gamma)q(x)} \right], \tag{25}$$

*where the densities $p(\theta)$, $p(\gamma)$, and $p(x \mid \theta)$ are exponential families and $p(\theta)$ is the natural exponential family conjugate prior to $p(x \mid \theta)$, as in Eqs. (18)-(19). Given a parameterization of the variational factors as*

$$q(\theta) = \exp\left\{ \langle \eta_\theta, \ t_\theta(\theta) \rangle - \log Z_\theta(\eta_\theta) \right\}, \quad q(\gamma) = \exp\left\{ \langle \eta_\gamma, \ t_\gamma(\gamma) \rangle - \log Z_\gamma(\eta_\gamma) \right\},$$
$$q(x) = \exp\left\{ \langle \eta_x, \ t_x(x) \rangle - \log Z_x(\eta_x) \right\},$$

*let $\mathcal{L}(\eta_\theta, \eta_\gamma, \eta_x)$ denote the mean field variational inference objective Eq. (25) as a function of these variational parameters. We define the* SVAE *objective as*

$$\mathcal{L}_{\text{SVAE}}(\eta_\theta, \eta_\gamma, \phi) \triangleq \mathcal{L}(\eta_\theta, \eta_\gamma, \eta_x^*(\eta_\theta, \phi)),$$

*where $\eta_x^*(\eta_\theta, \phi)$ is defined as a local partial optimizer of the surrogate objective $\widehat{\mathcal{L}}$,*

$$\eta_x^*(\eta_\theta, \phi) \triangleq \arg\max_{\eta_x} \widehat{\mathcal{L}}(\eta_\theta, \eta_x^*(\eta_\theta, \phi), \phi),$$

*where the surrogate objective $\widehat{\mathcal{L}}$ is defined as*

$$\widehat{\mathcal{L}}(\eta_\theta, \eta_x, \phi) \triangleq \mathbb{E}_{q(\theta)q(x)} \left[ \log \frac{p(\theta)p(x \mid \theta)\exp\{\psi(x; y, \phi)\}}{q(\theta)q(x)} \right],$$
$$\psi(x; y, \phi) \triangleq \langle r(y; \phi), \ t_x(x) \rangle,$$

*for some* recognition model $r(y; \phi)$ *parameterized by $\phi \in \mathbb{R}^m$.*

The SVAE objective $\mathcal{L}_{\text{SVAE}}$ is a lower-bound for the partially-optimized mean field variational inference objective in the following sense.

**Proposition D.2** (The SVAE objective lower-bounds the mean field objective)
*The SVAE objective function $\mathcal{L}_{\text{SVAE}}$ lower-bounds the partially-optimized mean field objective $\mathcal{L}$ in the sense that*

$$\max_{q(x)} \mathcal{L}[\, q(\theta)q(\gamma)q(x) \,] \geq \max_{\eta_x} \mathcal{L}(\eta_\theta, \eta_\gamma, \eta_x) \geq \mathcal{L}_{\text{SVAE}}(\eta_\theta, \eta_\gamma, \phi) \quad \forall \phi \in \mathbb{R}^m,$$

*for any choice of function class $\{r(y;\phi)\}_{\phi\in\mathbb{R}^m}$ in Eq. (23). Furthermore, if there is some $\phi^*\in\mathbb{R}^m$ such that*

$$\psi(x;y,\phi^*)=\mathbb{E}_{q(\gamma)}\log p(y\,|\,x,\gamma)$$

*then the bound can be made tight in the sense that*

$$\max_{q(x)}\mathcal{L}[\,q(\theta)q(\gamma)q(x)\,]=\max_{\eta_x}\mathcal{L}(\eta_\theta,\eta_\gamma,\eta_x)=\max_{\phi}\mathcal{L}_{\mathrm{SVAE}}(\eta_\theta,\eta_\gamma,\phi).$$

*Proof.* The inequalities follow from the variational principle and the definition of the SVAE objective $\mathcal{L}_{\mathrm{SVAE}}$. In particular, by Lemma C.2 the optimal factor over all possible densities is given by

$$q^{**}(x)\propto\exp\left\{\langle\mathbb{E}_{q(\theta)}\eta_x^0(\theta),\,t_x(x)\rangle+\mathbb{E}_{q(\gamma)}\log p(y\,|\,x,\gamma)\right\},\tag{26}$$

while we restrict the factor $q(x)$ to have a particular exponential family form indexed by parameter $\eta_x$, namely $q(x)\propto\exp\left\{\langle\eta_x,\,t_x(x)\rangle\right\}$. In the definition of $\mathcal{L}_{\mathrm{SVAE}}$ we also restrict the parameter $\eta_x$ to be set to $\eta_x^*(\eta_\theta,\phi)$, a particular function of $\eta_\theta$ and $\phi$, rather than setting it to the value that maximizes the mean field objective $\mathcal{L}$. Finally, equality holds when we can set $\phi$ to match the optimal $\eta_x$ and that choice yields the optimal factor given in Eq. (26). $\square$

Proposition D.2 motivates the SVAE optimization problem: by using gradient-based optimization to maximize $\mathcal{L}_{\mathrm{SVAE}}(\eta_\theta,\eta_\gamma,\phi)$ we are maximizing a lower-bound on the model evidence $\log p(y)$ and correspondingly minimizing the KL divergence from our variational family to the target posterior. Furthermore, it motivates choosing the recognition model function class $\{r(y;\phi)\}_{\phi\in\mathbb{R}^m}$ to be as rich as possible.

As we show in the following, choosing $\eta_x^*(\eta_\theta,\phi)$ to be a local partial optimizer of the surrogate objective $\widehat{\mathcal{L}}$ provides two significant computational advantages. First, it allows us to provide a simple expression for an unbiased estimate of the natural gradient $\widetilde{\nabla}_{\eta_\theta}\mathcal{L}_{\mathrm{SVAE}}$, as we describe next in Section D.2. Second, it allows $\eta_x^*(\eta_\theta,\phi)$ to be computed efficiently by exploiting exponential family structure, as we show in Section D.4.

### D.2 Estimating the natural gradient $\widetilde{\nabla}_{\eta_\theta}\mathcal{L}_{\mathrm{SVAE}}$

The definition of $\eta_x^*$ in terms of the surrogate objective $\widehat{\mathcal{L}}$ enables a simple and computationally efficient expression for the natural gradient with respect to the conjugate global variational parameters, $\widetilde{\nabla}_{\eta_\theta}\mathcal{L}_{\mathrm{SVAE}}(\eta_\theta,\eta_\gamma,\phi)$, as we show in the next proposition.

**Proposition D.3** (Natural gradient of the SVAE objective)
*The natural gradient of the SVAE objective Eq. (24) with respect to the conjugate global variational parameters $\eta_\theta$ is*

$$\widetilde{\nabla}_{\eta_\theta}\mathcal{L}_{\mathrm{SVAE}}(\eta_\theta,\eta_\gamma,\phi)=\left(\eta_\theta^0+\mathbb{E}_{q^*(x)}\left[(t_x(x),1)\right]-\eta_\theta\right)+\left(\nabla_{\eta_x}\mathcal{L}(\eta_\theta,\eta_\gamma,\eta_x^*(\eta_\theta,\phi)),0\right)$$

*where the first term is the SVI natural gradient from Corollary C.4, using*

$$q^*(x)\triangleq\exp\left\{\langle\eta_x^*(\eta_\theta,\phi),\,t_x(x)\rangle-\log Z_x(\eta_x^*(\eta_\theta,\phi))\right\},$$

*and where a stochastic estimate of the second term is computed as part of the backward pass for the gradient $\nabla_\phi\mathcal{L}(\eta_\theta,\eta_\gamma,\eta_x^*(\eta_\theta,\phi))$.*

*Proof.* First we use the chain rule, analogously to Eq. (1), to write the gradient as

$$\nabla_{\eta_\theta}\mathcal{L}_{\mathrm{SVAE}}(\eta_\theta,\eta_\gamma,\phi)=\left(\nabla^2\log Z_\theta(\eta_\theta)\right)\left(\eta_\theta^0+\mathbb{E}_{q^*(x)}\left[(t_{xy}(x,y),1)\right]-\eta_\theta\right)$$
$$+\left(\nabla_{\eta_\theta}\eta_x^*(\eta_\theta,\phi)\right)\left(\nabla_{\eta_x}\mathcal{L}(\eta_\theta,\eta_\gamma,\eta_x^*(\eta_\theta,\phi))\right),\tag{27}$$

where the first term is the same as the SVI gradient derived in Proposition C.3. In the case of SVI, the second term is zero because $\eta_x^*$ is chosen as a partial optimizer of $\mathcal{L}$, but for the SVAE objective the second term is nonzero in general, and the remainder of this proof amounts to deriving a simple expression for it.

We compute the term $\nabla_{\eta_\theta} \eta_x^*(\eta_\theta, \phi)$ in Eq. (27) in terms of the gradients of the surrogate objective $\widehat{\mathcal{L}}$ using the Implicit Function Theorem given in Corollary A.5, which yields

$$\nabla_{\eta_\theta} \eta_x^*(\eta_\theta, \phi) = -\nabla_{\eta_\theta \eta_x}^2 \widehat{\mathcal{L}}(\eta_\theta, \eta_x^*(\eta_\theta, \phi), \phi) \left( \nabla_{\eta_x \eta_x}^2 \widehat{\mathcal{L}}(\eta_\theta, \eta_x^*(\eta_\theta, \phi), \phi) \right)^{-1}. \tag{28}$$

First, we compute the gradient of $\widehat{\mathcal{L}}$ with respect to $\eta_x$, writing

$$\begin{aligned}
\nabla_{\eta_x} \widehat{\mathcal{L}}(\eta_\theta, \eta_x, \phi) &= \nabla_{\eta_x} \left[ \mathbb{E}_{q(\theta)q(x)} \left[ \log \frac{p(x \mid \theta) \exp\{\psi(x; y, \phi)\}}{q(x)} \right] \right] \\
&= \nabla_{\eta_x} \left[ \langle \mathbb{E}_{q(\theta)} \eta_x^0(\theta) + r(y; \phi) - \eta_x, \nabla \log Z_x(\eta_x) \rangle + \log Z_x(\eta_x) \right] \\
&= \left( \nabla^2 \log Z_x(\eta_x) \right) \left( \mathbb{E}_{q(\theta)} \eta_x^0(\theta) + r(y; \phi) - \eta_x \right). \tag{29}
\end{aligned}$$

Thus as a consequence of the first-order stationary condition $\nabla_{\eta_x} \widehat{\mathcal{L}}(\eta_\theta, \eta_x^*(\eta_\theta, \phi), \phi) = 0$ and the fact that $\nabla^2 \log Z_x(\eta_x)$ is always positive definite for minimal exponential families, we have

$$\mathbb{E}_{q(\theta)} \eta_x^0(\theta) + r(y; \phi) - \eta_x^*(\eta_\theta, \phi) = 0, \tag{30}$$

which is useful in simplifying the expressions to follow.

Continuing with the calculation of the terms in Eq. (28), we compute $\nabla_{\eta_x \eta_x}^2 \widehat{\mathcal{L}}$ by differentiating the expression in Eq. (29) again, writing

$$\begin{aligned}
\nabla_{\eta_x \eta_x}^2 \widehat{\mathcal{L}}(\eta_\theta, \eta_x^*(\eta_\theta, \phi), \phi) &= -\nabla^2 \log Z_x(\eta_x^*(\eta_\theta, \phi)) \tag{31} \\
&\quad + \left( \nabla^3 \log Z_x(\eta_x^*(\eta_\theta, \phi)) \right) \left( \mathbb{E}_{q(\theta)} \eta_x^0(\theta) + r(y; \phi) - \eta_x^*(\eta_\theta, \phi) \right) \\
&= -\nabla^2 \log Z_x(\eta_x^*(\eta_\theta, \phi)),
\end{aligned}$$

where the last line follows from using the first-order stationary condition Eq. (30). Next, we compute the other term $\nabla_{\eta_\theta \eta_x}^2 \widehat{\mathcal{L}}$ by differentiating Eq. (29) with respect to $\eta_\theta$ to yield

$$\nabla_{\eta_\theta \eta_x}^2 \widehat{\mathcal{L}}(\eta_\theta, \eta_x^*(\eta_\theta, \phi), \phi) = \left( \nabla^2 \log Z_\theta(\eta_\theta) \right) \begin{pmatrix} \nabla^2 \log Z_x(\eta_x^*(\eta_\theta, \phi)) \\ 0 \end{pmatrix},$$

where the latter matrix is $\nabla^2 \log Z_x(\eta_x^*(\eta_\theta, \phi))$ padded by a row of zeros.

Plugging these expressions back into Eq. (28) and cancelling, we arrive at

$$\nabla_{\eta_\theta} \eta_x^*(\eta_\theta, \phi) = \nabla^2 \log Z_\theta(\eta_\theta) \begin{pmatrix} I \\ 0 \end{pmatrix},$$

and so we have an expression for the gradient of the SVAE objective as

$$\begin{aligned}
\nabla_{\eta_\theta} \mathcal{L}_{\text{SVAE}}(\eta_\theta, \eta_\gamma, \phi) &= \left( \nabla^2 \log Z_\theta(\eta_\theta) \right) \left( \eta_\theta^0 + \mathbb{E}_{q^*(x)} \left[ (t_{xy}(x, y), 1) \right] - \eta_\theta \right) \\
&\quad + \left( \nabla^2 \log Z_\theta(\eta_\theta) \right) \left( \nabla_{\eta_x} \mathcal{L}(\eta_\theta, \eta_\gamma, \eta_x^*(\eta_\theta, \phi)), 0 \right).
\end{aligned}$$

When we compute the natural gradient, the Fisher information matrix factors on the left of each term cancel, yielding the result in the proposition. $\qquad\square$

As a consequence of Proposition D.3, the SVAE algorithm is almost as simple as the SVI algorithm, which applies only to complete-data conjugate models, yet can handle general likelihood densities $p(y \mid x, \gamma)$. In particular, to compute the natural gradient $\widetilde{\nabla}_{\eta_\theta} \mathcal{L}_{\text{SVAE}}$ we only need to compute the same expected sufficient statistics as in the SVI algorithm plus a correction term that, as we show next, is already estimated via automatic differentiation as part of estimating the gradient $\nabla_\phi \mathcal{L}_{\text{SVAE}}$.

The proof of Proposition D.3 used the necessary condition for unconstrained local optimality to simplify the expression in Eq. (31). This simplification does not necessarily hold if $\eta_x$ is constrained; for example, if the factor $q(x)$ has additional factorization structure that is not present in $p(x \mid \theta)$, that additional structure can manifest as linear constraints on the natural parameter $\eta_x$. In the cases we consider here and in Section D.4 the factorization structure in the variational family matches that in the model and so the stationarity conditions apply. Note also that for Gaussian $q(x)$ the same simplification always applies because third and higher-order cumulants are zero for Gaussians and hence $\nabla^3 \log Z_x(\eta_x) = 0$.

**D.3 Estimating the gradients $\nabla_\phi \mathcal{L}_{\mathrm{SVAE}}$ and $\nabla_{\eta_\gamma} \mathcal{L}_{\mathrm{SVAE}}$**

To compute an unbiased stochastic estimate of the gradients $\nabla_\phi \mathcal{L}_{\mathrm{SVAE}}(\eta_\theta, \eta_\gamma, \phi)$ and $\nabla_{\eta_\gamma} \mathcal{L}_{\mathrm{SVAE}}(\eta_\theta, \eta_\gamma, \phi)$ we use the reparameterization trick [11], which is simply to differentiate a stochastic estimate of the objective $\mathcal{L}_{\mathrm{SVAE}}(\eta_\theta, \eta_\gamma, \phi)$ as a function of $\phi$ and $\eta_\gamma$. To isolate the terms that require this sample-based approximation from those that can be computed directly, we rewrite the objective as

$$\mathcal{L}_{\mathrm{SVAE}}(\eta_\theta, \eta_\gamma, \phi) = \mathbb{E}_{q(\gamma)q^*(x)} \log p(y \,|\, x, \gamma) - \mathrm{KL}(q(\theta)q(\gamma)q^*(x) \,\|\, p(\theta, \gamma, x)) \qquad (32)$$

where, as before,

$$q^*(x) \triangleq \exp\left\{ \langle \eta_x^*(\eta_\theta, \phi), \ t_x(x) \rangle - \log Z_x(\eta_x^*(\eta_\theta, \phi)) \right\}$$

and so the dependence of the expression in Eq. (32) on $\phi$ is through $\eta_x^*(\eta_\theta, \phi)$.

Only the first term in Eq. (32) needs to be estimated with the reparameterization trick. Due to the exponential family structure, the second term in Eq. (32) has a simple expression, as we show in the following proposition.

**Proposition D.4** (Computing $\mathrm{KL}(q(\theta)q(\gamma)q^*(x) \,\|\, p(\theta, \gamma, x))$)
*The KL divergence term in the SVAE objective in Eq. (32) can be computed as*

$$\begin{aligned}
\mathrm{KL}(q(\theta)q(\gamma)q^*(x) \,\|\, p(\theta, \gamma, x)) = {}& \mathrm{KL}(q(\theta) \,\|\, p(\theta)) + \mathrm{KL}(q(\gamma) \,\|\, p(\gamma)) \\
& + \log Z_x(\eta_x^*(\eta_\theta, \phi)) - \langle r(y; \phi), \, \nabla \log Z_x(\eta_x^*(\eta_\theta, \phi)) \rangle,
\end{aligned}$$

*where the first two terms can be computed with Eq. (5) of Proposition B.7.*

*Proof.* Using basic properties of the KL divergence for factorized densities we have

$$\begin{aligned}
\mathrm{KL}(q(\theta)q(\gamma)q^*(x) \,\|\, p(\theta, \gamma, x)) = {}& \mathrm{KL}(q(\theta) \,\|\, p(\theta)) + \mathrm{KL}(q(\gamma) \,\|\, p(\gamma)) \\
& + \mathbb{E}_{q(\theta)} \mathrm{KL}(q^*(x) \,\|\, p(x \,|\, \theta)),
\end{aligned}$$

and so it remains to compute an expression for the final term. Using Proposition B.7 and the stationarity condition given in Eq. (30), we have

$$\begin{aligned}
\mathbb{E}_{q(\theta)} \mathrm{KL}(q^*(x) \,\|\, p(x \,|\, \theta)) = {}& \langle \mathbb{E}_{q(\theta)} \eta_x^0(\theta) - \eta_x^*(\eta_\theta, \phi), \, \nabla \log Z_x(\eta_x^*(\eta_\theta, \phi)) \rangle \\
& + \log Z_x(\eta_x^*(\eta_\theta, \phi)) \\
= {}& \log Z_x(\eta_x^*(\eta_\theta, \phi)) - \langle r(y; \phi), \, \nabla \log Z_x(\eta_x^*(\eta_\theta, \phi)) \rangle.
\end{aligned}$$

Adding these terms yields the desired result. $\qquad\square$

Proposition D.4 gives an explicit formula for computing the KL divergence term involving only the log partition functions of the variational exponential families and their gradients. When the variational family is chosen so that these log partition functions can be computed efficiently, the gradients of this term with respect to $\phi$ and $\eta_\gamma$ can also be computed efficiently by reverse-mode automatic differentiation.

We summarize the results of this subsection in the following proposition.

**Proposition D.5** (Estimating $\nabla_\phi \mathcal{L}_{\mathrm{SVAE}}$ and $\nabla_{\eta_\gamma} \mathcal{L}_{\mathrm{SVAE}}$)
*Let $\hat{\gamma}(\eta_\gamma) \sim q(\gamma)$ and $\hat{x}(\phi) \sim q^*(x)$ be samples of $q(\gamma)$ and $q^*(x)$, respectively. Unbiased estimates of the gradients $\nabla_\phi \mathcal{L}_{\mathrm{SVAE}}(\eta_\theta, \eta_\gamma, \phi)$ and $\nabla_{\eta_\gamma} \mathcal{L}_{\mathrm{SVAE}}(\eta_\theta, \eta_\gamma, \phi)$ are given by*

$$\begin{aligned}
\nabla_\phi \mathcal{L}_{\mathrm{SVAE}}(\eta_\theta, \eta_\gamma, \phi) &\approx \nabla_\phi \log p(y \,|\, \hat{x}(\phi), \hat{\gamma}(\eta_\gamma)) - \nabla_\phi \mathrm{KL}(q(\theta)q^*(x) \,\|\, p(\theta, x)), \\
\nabla_{\eta_\gamma} \mathcal{L}_{\mathrm{SVAE}}(\eta_\theta, \eta_\gamma, \phi) &\approx \nabla_{\eta_\gamma} \log p(y \,|\, \hat{x}(\phi), \hat{\gamma}(\eta_\gamma)) - \nabla_{\eta_\gamma} \mathrm{KL}(q(\gamma) \,\|\, p(\gamma)).
\end{aligned}$$

*Both of these gradients can be computed by automatically differentiating the Monte Carlo estimate of $\mathcal{L}_{\mathrm{SVAE}}$ given by*

$$\mathcal{L}_{\mathrm{SVAE}}(\eta_\theta, \eta_\gamma, \phi) \approx \log p(y \,|\, \hat{x}(\phi), \hat{\gamma}(\eta_\gamma)) - \mathrm{KL}(q(\theta)q(\gamma)q^*(x) \,\|\, p(\theta, \gamma, x))$$

*with respect to $\eta_\gamma$ and $\phi$, respectively, where the second term can be computed via Proposition D.4.*

## D.4 Partially optimizing $\widehat{\mathcal{L}}$ using conjugacy structure

In Section D.1 we defined the SVAE objective in terms of a function $\eta_x^*(\eta_\theta, \phi)$, which was itself implicitly defined in terms of first-order stationary conditions for an auxiliary objective $\widehat{\mathcal{L}}(\eta_\theta, \eta_x, \phi)$. Here we show how $\widehat{\mathcal{L}}$ admits efficient local partial optimization in the same way as the conditionally conjugate model of Section C.4.

In this section we consider additional structure in the local latent variables. Specifically, as in Section C.4, we introduce to the notation another set of local latent variables $z$ in addition to the local latent variables $x$. However, unlike Section C.4, we still consider general likelihood families $p(y \,|\, x, \gamma)$.

Let $p(z, x \,|\, \theta)$ be an exponential family and $p(\theta)$ be its corresponding natural exponential family conjugate prior, writing

$$p(\theta) = \exp\left\{ \langle \eta_\theta^0, \, t_\theta(\theta) \rangle - \log Z_\theta(\eta_\theta^0) \right\}, \tag{33}$$

$$p(z, x \,|\, \theta) = \exp\left\{ \langle \eta_{zx}^0(\theta), \, t_{zx}(z, x) \rangle - \log Z_{zx}(\eta_{zx}^0(\theta)) \right\}$$

$$= \exp\left\{ \langle t_\theta(\theta), \, (t_{zx}(z, x), 1) \rangle \right\}$$

where we have used $t_\theta(\theta) = \left( \eta_{zx}^0(\theta), -\log Z_{zx}(\eta_{zx}^0(\theta)) \right)$ in Eq. (12). Additionally, let $t_{zx}(z, x)$ be a multilinear polynomial in the statistics $t_z(z)$ and $t_x(x)$, and let $p(z \,|\, \theta)$ and $p(x \,|\, z, \theta)$ be a conjugate pair of exponential families, writing

$$p(z \,|\, \theta) = \exp\left\{ \langle \eta_z^0(\theta), \, t_z(z) \rangle - \log Z_z(\eta_z^0(\theta)) \right\},$$

$$p(x \,|\, z, \theta) = \exp\left\{ \langle \eta_x^0(z, \theta), \, t_x(x) \rangle - \log Z_x(\eta_x^0(z, \theta)) \right\}$$

$$= \exp\left\{ \langle t_z(z), \eta_x^0(\theta)^{\mathsf{T}}(t_x(x), 1) \rangle \right\}.$$

Let $p(y \,|\, x, \gamma)$ be a general family of densities (not necessarily an exponential family) and let $p(\gamma)$ be an exponential family prior on its parameters of the form

$$p(\gamma) = \exp\left\{ \langle \eta_\gamma^0, \, t_\gamma(\gamma) \rangle - \log Z_\gamma(\eta_\gamma^0) \right\}.$$

The corresponding variational factors are

$$q(\theta) = \exp\left\{ \langle \eta_\theta, \, t_\theta(\theta) \rangle - \log Z_\theta(\eta_\theta) \right\}, \qquad q(\gamma) = \exp\left\{ \langle \eta_\gamma, \, t_\gamma(\gamma) \rangle - \log Z_\gamma(\eta_\gamma) \right\},$$

$$q(z) = \exp\left\{ \langle \eta_z, \, t_z(z) \rangle - \log Z_z(\eta_z) \right\}, \qquad q(x) = \exp\left\{ \langle \eta_x, \, t_x(x) \rangle - \log Z_x(\eta_x) \right\}.$$

As in Section D.1, we construct the surrogate objective $\widehat{\mathcal{L}}$ to allow us to exploit exponential family and conjugacy structure. In particular, we construct $\widehat{\mathcal{L}}$ to resemble the mean field objective, namely

$$\mathcal{L}(\eta_\theta, \eta_\gamma, \eta_z, \eta_x) \triangleq \mathbb{E}_{q(\theta)q(\gamma)q(z)q(x)}\left[ \log \frac{p(\theta)p(\gamma)p(z \,|\, \theta)p(x \,|\, z, \theta)p(y \,|\, x, \gamma)}{q(\theta)q(\gamma)q(z)q(x)} \right],$$

but in $\widehat{\mathcal{L}}$ we replace the $\log p(y \,|\, x, \gamma)$ likelihood term, which may be a general family of densities without much structure, with a more tractable approximation,

$$\widehat{\mathcal{L}}(\eta_\theta, \eta_z, \eta_x, \phi) \triangleq \mathbb{E}_{q(\theta)q(z)q(x)}\left[ \log \frac{p(\theta)p(z \,|\, \theta)p(x \,|\, z, \theta)\exp\{\psi(x; y, \phi)\}}{q(\theta)q(z)q(x)} \right],$$

where $\psi(x; y, \phi)$ is a function on $x$ that resembles a conjugate likelihood for $p(x \,|\, z, \theta)$,

$$\psi(x; y, \phi) \triangleq \langle r(y; \phi), \, t_x(x) \rangle, \qquad \phi \in \mathbb{R}^m.$$

We then define $\eta_z^*(\eta_\theta, \phi)$ and $\eta_x^*(\eta_\theta, \phi)$ to be local partial optimizers of $\widehat{\mathcal{L}}$ given fixed values of the other parameters $\eta_\theta$ and $\phi$, and in particular they satisfy the first-order necessary optimality conditions

$$\nabla_{\eta_z} \widehat{\mathcal{L}}(\eta_\theta, \eta_z^*(\eta_\theta, \phi), \eta_x^*(\eta_\theta, \phi), \phi) = 0, \qquad \nabla_{\eta_x} \widehat{\mathcal{L}}(\eta_\theta, \eta_z^*(\eta_\theta, \phi), \eta_x^*(\eta_\theta, \phi), \phi) = 0.$$

The SVAE objective is then

$$\mathcal{L}_{\mathrm{SVAE}}(\eta_\theta, \eta_\gamma, \phi) \triangleq \mathcal{L}(\eta_\theta, \eta_\gamma, \eta_z^*(\eta_\theta, \phi), \eta_x^*(\eta_\theta, \phi)). \tag{34}$$

The structure of the surrogate objective $\widehat{\mathcal{L}}$ is chosen so that it resembles the mean field variational inference objective for the conditionally conjugate model of Section C.4, and as a result we can use the same block coordinate ascent algorithm to efficiently find partial optimizers $\eta_z^*(\eta_\theta, \phi)$ and $\eta_x^*(\eta_\theta, \phi)$.

**Proposition D.6** (Computing $\eta_z^*(\eta_\theta, \phi)$ and $\eta_x^*(\eta_\theta, \phi)$)

*Let the densities $p(\theta, \gamma, z, x, y)$ and $q(\theta)q(\gamma)q(z)q(x)$ and the objectives $\mathcal{L}$, $\widehat{\mathcal{L}}$, and $\mathcal{L}_{\mathrm{SVAE}}$ be as in Eqs. (33)-(34). The partial optimizers $\eta_z^*$ and $\eta_x^*$, defined by*

$$\eta_z^* \triangleq \arg\max_{\eta_z} \widehat{\mathcal{L}}(\eta_\theta, \eta_z, \eta_x, \phi), \qquad\qquad \eta_x^* \triangleq \arg\max_{\eta_x} \widehat{\mathcal{L}}(\eta_\theta, \eta_z, \eta_x, \phi)$$

*with the other arguments fixed, are are given by*

$$\eta_z^* = \mathbb{E}_{q(\theta)}\eta_z^0(\theta) + \mathbb{E}_{q(\theta)q(x)}\eta_x^0(\theta)^{\mathsf{T}}(t_x(x), 1), \qquad \eta_x^* = \mathbb{E}_{q(\theta)q(z)}\eta_x^0(z, \theta) + r(y; \phi), \quad (35)$$

*and by alternating the expressions in Eq. (35) as updates we can compute $\eta_z^*(\eta_\theta, \phi)$ and $\eta_x^*(\eta_\theta, \phi)$ as local partial optimizers of $\widehat{\mathcal{L}}$.*

*Proof.* These updates follow immediately from Lemma C.2. Note in particular that the stationary conditions $\nabla_{\eta_z}\widehat{\mathcal{L}} = 0$ and $\nabla_{\eta_x}\widehat{\mathcal{L}} = 0$ yield the each expression in Eq. (35), respectively. □

The other properties developed in Propositions D.2, D.3, and D.5 also hold true for this model because it is a special case in which we have separated out the local variables, denoted $x$ in earlier sections, into two groups, denoted $z$ and $x$ here, to match the exponential family structure in $p(z \mid \theta)$ and $p(x \mid z, \theta)$, and performed unconstrained optimization in each of the variational parameters. However, the expression for the natural gradient is slightly simpler for this model than the corresponding version of Proposition D.3. For completeness, we restate Proposition D.3 using the notation of this section.

**Proposition D.7** (Natural gradient of the SVAE objective)

*The natural gradient of the SVAE objective Eq. (34) with respect to the conjugate global variational parameters $\eta_\theta$ is*

$$\widetilde{\nabla}_{\eta_\theta}\mathcal{L}_{\mathrm{SVAE}}(\eta_\theta, \eta_\gamma, \phi) = \left(\eta_\theta^0 + \mathbb{E}_{q^*(z)q^*(x)}\left[(t_{zx}(z, x), 1)\right] - \eta_\theta\right)$$
$$+ \left(\nabla_{\eta_x}\mathcal{L}(\eta_\theta, \eta_\gamma, \eta_z^*(\eta_\theta, \phi), \eta_x^*(\eta_\theta, \phi)), 0\right).$$

*Proof.* Note that the optimality condition satisfied by $\eta_z^*$, namely

$$\nabla_{\eta_z}\widehat{\mathcal{L}}(\eta_\theta, \eta_z^*(\eta_\theta, \phi), \eta_x^*(\eta_\theta, \phi), \phi) = \nabla_{\eta_z}\mathbb{E}_{q(\theta)q(z)q(x)}\left[\log \frac{p(z \mid \theta)p(x \mid z, \theta)}{q(z)}\right] = 0,$$

also implies that it is stationary for $\mathcal{L}$,

$$\nabla_{\eta_z}\mathcal{L}(\eta_\theta, \eta_\gamma, \eta_z^*(\eta_\theta, \phi), \eta_x^*(\eta_\theta, \phi)) = \nabla_{\eta_z}\mathbb{E}_{q(\theta)q(z)q(x)}\left[\log \frac{p(z \mid \theta)p(x \mid z, \theta)}{q(z)}\right] = 0,$$

and so by Proposition A.3 the term involving $\nabla_{\eta_z}\mathcal{L}$ does not appear in the chain rule expansion for the gradient $\nabla_{\eta_z}\mathcal{L}_{\mathrm{SVAE}}(\eta_\theta, \eta_\gamma, \phi)$. The remainder of the proof follows that of Proposition D.3. □

# E  Experiment details and expanded figures

For the synthetic 1D dot video data, we trained an LDS SVAE on 80 random image sequences each of length 50, using one sequence per update, and show the model's future predictions given a prefix of a longer sequence. We used MLP image and recognition models each with one hidden layer of 50 units and a latent state space of dimension 8. The middle and bottom panels of Fig. 1 show the model's predictions and sampled latent state trajectories, respectively, with the predictions conditioned on the data up to the vertical red line. The model is able both to represent the image accurately and to make long-term predictions while modeling uncertainty.

See also videos of training a warped mixture and training a nonlinear LDS.

(a) Predictions after 200 training steps.　　　(b) Predictions after 1100 training steps.

Figure 1: Predictions from an LDS SVAE fit to 1D dot image data at two stages of training. The top panel shows an example sequence with time on the horizontal axis. The middle panel shows the noiseless predictions given data up to the vertical line, while the bottom panel shows the latent states.

(a) Comparing natural (blue) and standard (orange) gradient updates. The X's mark early termination due to indefiniteness.

(b) Random 2D subspace of MLP observation model fit to mouse data.

Figure 2: Panel (a) compares natural and standard gradient updates and Panel (b) shows a random 2D subspace in the image manifold coordinates learned by fitting a VAE to mouse depth video data.

Figure 3: Examples of predictions from an LDS SVAE fit to depth video. In each panel, the top row is a sampled prediction from the LDS SVAE and the bottom row is real data. To the left of the line, the model is conditioned on the corresponding data frames and hence generates denoised versions of the same images. To the right of the line, the model is not conditioned on the data, thus illustrating the model's predictions. The frame sequences are temporally subsampled to reduce their length, showing one of every four video frames.

(a) Beginning a rear

(b) Grooming

(c) Extension into running

(d) Fall from rear

Figure 4: Examples of behavior states inferred from depth video. For each state, four example frame sequences are shown, including frames during which the given state was most probable according to the variational distribution on the hidden state sequence. Each frame sequence is padded on both sides, with a square in the lower-right of a frame depicting that the state was active in that frame. The frame sequences are temporally subsampled to reduce their length, showing one of every four video frames. Examples were chosen to have durations close to the median duration for that state.

## Footnotes

[1]For a discussion of differentiability issues when there is more than one optimizer, i.e. when $\arg\max_y f(x, y)$ has more than one element, see Danskin [2], Fiacco [3, Section 2.4], and Bonnans et al. [4, Chapter 4]. Here we only consider the sensitivity of local stationary points and assume differentiability almost everywhere.

[2]Families that are not minimal, like the density of the categorical distribution, can be treated by restricting all algebraic operations to the subspace spanned by the statistic, i.e. to the smallest $V \subset \mathbb{R}^n$ with range $t_x \subseteq V$.

[3]The parametric form for $q(x)$ need not be restricted a priori, but rather without loss of generality given the surrogate objective Eq. (22) and the form of $\psi$ used in Eq. (23), the optimal factor $q(x)$ is in the same family as $p(x \mid \theta)$. We treat it as a restriction here so that we can proceed with more concrete notation.