[Reviews · NeurIPS 2016]

Reviewer 1

Summary

Arguing that graphical models and neural networks have complementary strengths, the authors introduce a family of probabilistic models that combine structured prior distributions formulated as graphical models with highly nonlinear observation models implemented using neural networks. The goal is to combine the interpretability and efficient inference algorithms of graphical models with the representation learning power of neural nets. The main contribution of the paper is an efficient stochastic variational inference algorithm for training such hybrid models that uses a recognition model, implemented a neural network, to deal with the non-conjugate observation model to enable efficient mean field updates for inferring the local latent variables. The experimental section is minimal, showing qualitative results on a synthetic dataset and a small dataset of low-resolution video.

Qualitative Assessment

The idea of using a neural recognition model to implement conjugacy in a surrogate objective used to infer the local latent variables seems like a powerful and important one. At the same time it is less direct than taking the variational autoencoder (VAE) route and using the recognition model to produce the parameters of the variational posterior directly. It would have been good to contrast the two approaches and discuss their relative strengths and weaknesses. I found the paper to be well-written overall, though it was too dense and not detailed enough in parts. The supplementary material was essential for me to understand some details unclear from the main text. Though the experimental section is very bare-bones and lacks quantitative results, I do not think that is a serious weakness for a paper with a substantial conceptual contribution like this one. The related work section missed several recent papers on sequence modeling in the VAE framework. Another notable omission was the work of Titsias and Lazaro-Gredilla [1]. References [1] M. K. Titsias and M. Lazaro-Gredilla. Doubly Stochastic Variational Bayes for non-Conjugate Inference, ICML, 2014

Confidence in this Review

2-Confident (read it all; understood it all reasonably well)


Reviewer 2

Summary

The proposed approach essentially introduces a finite mixture model, where each component distribution is a Gaussian latent variable model, the likelihood means and covariances of which are parameterized via neural networks.

Qualitative Assessment

The main concept of the paper is not different from existing deep generative model (DGM) formulations where the postulated likelihood is a finite mixture of Gaussians. What differentiates this work from existing DGMs concerns the postulated latent variable posteriors: Instead of considering a Gaussian posterior (conditional on the mixture component) that is parameterized via deep neural networks, the authors consider here a conventional formulation, where a set of Normal-Wishart hyper-priors is further imposed over the Gaussian posterior mean and precision matrix. I have to admit that I’ve failed to understand why the authors drop variational posterior amortization to opt for this more simplistic solution. Using amortized variational posteriors is now the state-of-the-art approach in the literature, and is proven to allow for better modeling and inferential performance. The authors should have motivated this selection a lot more extensively. The presented experiments fail to provide substantial empirical evidence to support the efficacy and the usefulness of the approach. 
They are both limited to some simulated datasets and not exhaustively compared to the state of the art.

Confidence in this Review

3-Expert (read the paper in detail, know the area, quite certain of my opinion)


Reviewer 3

Summary

This paper presents a general modeling and inference framework that combines the strengths of probabilistic graphical models and the flexibility of neural network models. Graphical models are used to represent structured probability distributions and variational autoencoder ideas are used to learn the nonlinear data manifold in an approach that is referred to as the structured variational autoencoder (SVAE). A major challenge of hybrid models is that inference is often difficult. The main idea of this paper is to learn recognition networks that output conjugate graphical model potentials, allowing for the use of tractable graphical model inference algorithms. They outline an algorithm for doing inference in the SVAE model. The SVAE model is then applied to both a synthetic dataset and a real dataset. In the synthetic data experiment, a latent linear dynamical system (LDS) SVAE is trained to predict the trajectory of a dot that is bouncing back and forth in a sequence of images. The model seems to do a very good job on this relatively simple task. Next, they apply a LDS SVAE to model depth video recordings of mouse behavior, where the model seems to predict future frames relatively well. Finally, they apply a latent switching linear dynamical system (SLDS) SVAE model to try to learn discrete states corresponding to natural behavioral units.

Qualitative Assessment

Combining PGMs and deep learning/neural networks is a very active and promising area of research. This is an interesting paper, with it’s main contribution to the existing literature being that it presents a model that can account for discrete latent variables. This new capability suggests that it could be used in a variety of interesting applications, including the type of behavior representation modeling shown in one of the experiments. Overall, the organization of the paper is excellent, and the writing is clear. The technical part is however quite dense. Algorithm 1 in Section 4 is hard to follow. Some of the symbols in the algorithm don’t seem to be defined anywhere (and if they are, they’re too hard to find). One of the strengths of the paper is that the approach is quite general. The very general description of the method might make it hard to figure out how to apply the algorithm to a specific new model. It might be helpful to include in the appendix a detailed description of the algorithm applied to the examples in Figure 3. My main criticism is that there is no comparison to existing baselines, except for the "toy" example in Figure 1. I do believe that there are advantages to "structured" representations (at the very least, in terms of interpretability), but it would have been nice to see the results achieved by some baseline (existing) method. E.g., in Figure 5, it’s hard for someone who is unfamiliar with the data to determine if the model is making good predictions without having something to compare to. Section 6.3 and Figure 6 should be explained in more detail. It seems that one of the more interesting contributions of this approach is that the latent switching linear dynamical system (SLDS) model can identify discrete latent behavioral states that influence the observed dynamics. Without more detail, it’s very hard to tell what is going on. For example, Section 6.3 states that there are 30 discrete states, but Figure 6 only shows two of these - are the other states semantically meaningful as well, or were these two picked because they were the only useful ones?

Confidence in this Review

2-Confident (read it all; understood it all reasonably well)


Reviewer 4

Summary

The paper presents a method for generalizing variational autoencoders (VAEs) to structured graphical models with general, non-linear observation models. The main idea of the approach is to use the VAE framework to predict potential functions rather than the variational parameters themselves to facilitate inference on structured latent representations.

Qualitative Assessment

This is a good idea, and as far as I can ascertain, all claims (bounds, etc.) hold. However the manuscript in it's present form is a little hard to follow. For example The main objective, L_{SVAE}, is never even introduced in the main body of the text, and important proofs are scattered throughout the lengthy supplementary material---I think the authors can definitely improve the exposition substantially with some effort. A tighter correspondence around the algorithm description and an expanded exposition in the appendix might be useful where appropriate. Perhaps Figure 6 could go in the appendix to make some room for further explanations. The experimental results are proof of concept I suppose, but after getting excited about the new machinery, it was a let-down to see the model exercised only on toy data and mouse depth data, and compared to other methods only on synthetic spiral data. I hope the authors will be able to demonstrate their model on a more involved and well-known task if the paper gets accepted. Notes: - Appendix, eq (28) constant terms left in posterior.u - Sum-product networks represent discrete sub-structures and are efficient... line 86- z_n should be z_{n+1} according to Figure 2...

Confidence in this Review

2-Confident (read it all; understood it all reasonably well)


Reviewer 5

Summary

The authors provide an approach to learn models that combine nonlinear likelihoods from neural networks with structured latent variables. They bring together several pre-existing tools including stochastic variational inference, message passing, and backpropagation using the reparameterization trick. Efficiency of optimization is improved where possible by using conjugate exponential families and natural gradients. The paper is well written with helpful examples and comments on related work.

Qualitative Assessment

The paper is clearly written and neatly combines several earlier methods. To their credit, recent work that takes a similar approach [7,19,20,21,22] is cited and briefly described in Section 5. It would be worth adding a brief comment on Belanger and McCallum ICML 2016, Structured Prediction Energy Networks. Good examples are provided demonstrating scalability, and the videos are a nice bonus. Please could you clarify the strengths of new contributions here? My main concern is if bringing together this collection of earlier tools is a sufficient novel contribution for this conference. I do not feel strongly and am not an expert in this area.

Confidence in this Review

2-Confident (read it all; understood it all reasonably well)


Reviewer 6

Summary

This paper combines neural networks with graphical models. Variational inference and neural network learning are combined into a single optimization objective.

Qualitative Assessment

Let me start by acknowledging that I am not an expert on this topic, and I did not understand the details of this work. That being said, I found the contribution to be slightly incremental, or at least vague. I also feel like the carity of the presentation is lacking. The paper assumes that the reader is an expert on variational methods, and lots of arcane terminology is used.

Confidence in this Review

1-Less confident (might not have understood significant parts)